# MLL1 is required for PAX7 expression and satellite cell self-renewal in mice

Gregory C. Addicks [1,2,6], Caroline E. Brun[1,2,6], Marie-Claude Sincennes[1,2], John Saber[1,2], Christopher J. Porter[3], A. Francis Stewart[4], Patricia Ernst[5] & Michael A. Rudnicki [1,2]

PAX7 is a paired-homeobox transcription factor that specifies the myogenic identity of muscle stem cells and acts as a nodal factor by stimulating proliferation while inhibiting differentiation. We previously found that PAX7 recruits the H3K4 methyltransferases MLL1/2 to epigenetically activate target genes. Here we report that in the absence of *Mll1*, myoblasts exhibit reduced H3K4me3 at both *Pax7* and *Myf5* promoters and reduced *Pax7* and *Myf5* expression. *Mll1*-deficient myoblasts fail to proliferate but retain their differentiation potential, while deletion of *Mll2* had no discernable effect. Re-expression of PAX7 in committed *Mll1* cKO myoblasts restored H3K4me3 enrichment at the *Myf5* promoter and *Myf5* expression. Deletion of *Mll1* in satellite cells reduced satellite cell proliferation and self-renewal, and significantly impaired skeletal muscle regeneration. *Pax7* expression was unaffected in quiescent satellite cells but was markedly downregulated following satellite cell activation. Therefore, MLL1 is required for PAX7 expression and satellite cell function in vivo. Furthermore, PAX7, but not MLL1, is required for *Myf5* transcriptional activation in committed myoblasts.

[1] Sprott Centre for Stem Cell Research, Regenerative Medicine Program, Ottawa Hospital Research Institute, Ottawa, ON K1H 8L6, Canada. [2] Department of Cellular and Molecular Medicine, Faculty of Medicine, University of Ottawa, Ottawa, ON K1H 8M5, Canada. [3] Sprott Centre for Stem Cell Research, Ottawa Bioinformatics Core Facility, Ottawa Hospital Research Institute, Ottawa, ON K1H 8L6, Canada. [4] Genomics, Center for Molecular and Cellular Bioengineering, Technische Universität Dresden, Tatzberg 47, Dresden 01307, Germany. [5] Department of Pediatrics and Pharmacology, University of Colorado/Anschutz Medical Campus, Aurora, CO 80045, USA. [6]These authors contributed equally: Gregory C. Addicks, Caroline E. Brun. Correspondence and requests for materials should be addressed to M.A.R. (email: mrudnicki@ohri.ca)

Skeletal muscle has a remarkable capacity of regeneration that relies on a population of muscle stem cells termed satellite cells[1]. In resting adult muscle, satellite cells are mainly quiescent, but are poised for activation, allowing for muscle repair upon injury[2]. Activated satellite stem cells undergo asymmetric cell divisions to generate myogenic progenitor cells that proliferate and eventually differentiate by either fusing into damaged myofibers or generating new myofibers[3]. In addition, a subset of activated satellite cells will withdraw from the cell cycle and return to quiescence to replenish the muscle stem cell reservoir. Satellite cell function is highly regulated by extrinsic growth factors from the muscle stem cell niche as well as intrinsic transcriptional factors such as the paired-homeobox transcription factor PAX7 and the myogenic regulatory factors (MRFs)[4].

PAX7 is expressed by all satellite cells and is required for satellite cell function. $Pax7^{-/-}$ mice exhibit smaller myofibers at birth and fail to form a functional muscle stem cell reservoir, leading to impaired muscle regeneration and juvenile mortality[5–8]. Furthermore, conditional $Pax7$ deletion in adult mice strongly impairs the regenerative capacity of skeletal muscle due to proliferation defects and precocious differentiation of satellite cells, confirming the absolute requirement of PAX7 for satellite cell function[9,10]. Chromatin immunoprecipitation (ChIP)-sequencing in primary myoblasts revealed that PAX7 acts as a nodal factor by activating target genes involved in establishing myogenic identity and in stimulating proliferation while inhibiting differentiation[11,12]. A major target gene of PAX7 is $Myf5$, and loss of PAX7 significantly decreases $Myf5$ expression in both satellite cells and cultured myoblasts[6,9,11,13].

We previously showed that $Myf5$ transcriptional regulation by PAX7 involves the recruitment of the Trithorax complex, encompassing ASH2L, WDR5, RBBP5, and MLL1/2, to $Myf5$ regulatory sequences through direct interaction between PAX7 and MLL1/2[13,14]. MLL1/2 are closely related members of a family of six histone methyltransferases that specifically methylate histone H3 lysine 4 to an activating trimethyl state (H3K4me3)[15,16]. The ability of PAX7 to recruit this Trithorax complex to chromatin is regulated by CARM1 and p38γ MAPK[14,17]. Consistent with these findings, the $Myf5$ locus exhibits H3K4me3 enrichment around the transcription start site (TSS) in quiescent satellite cells as well as in primary myoblasts[12,13,18].

In this study, we uncover a specific role of MLL1 in regulating PAX7 expression, elucidating their involvement in $Myf5$ transcriptional control. Using conditional alleles of $Mll1$ and $Mll2$, we find that MLL1 directly regulates $Pax7$. In the absence of $Mll1$, myoblasts display reduced H3K4me3 enrichment at both $Pax7$ and $Myf5$ loci associated with the loss of $Pax7$ and $Myf5$ expression. As a consequence, $Mll1$-deficient myoblasts fail to proliferate but retain their potential to differentiate. By contrast, deletion of $Mll2$ has no apparent effect. Re-establishing PAX7 expression in committed $Mll1$ cKO myoblasts is sufficient to restore H3K4me3 enrichment at the $Myf5$ promoter and rescue $Myf5$ levels, indicating that PAX7, but not MLL1, is required for $Myf5$ expression in committed myoblasts. Finally, conditional $Mll1$ deletion in satellite cells dramatically impairs satellite cell proliferation, self-renewal, and skeletal muscle regeneration. Altogether, our data demonstrate that MLL1 is absolutely required for PAX7 expression and satellite cell function.

## Results

**Loss of MLL1 impairs $Pax7$ and $Myf5$ expression.** Both MLL1 ($Kmt2a$) and MLL2 ($Kmt2b$) can interact with PAX7 to activate $Myf5$ expression[13,14], suggesting that MLL1 and MLL2 may be functionally redundant. Thus, we investigated whether MLL1 and MLL2 share the same function in regulating $Myf5$ expression

using $Mll1$ and $Mll2$ conditional knockout myoblasts ($Mll1$ cKO and $Mll2$ cKO) and $Mll1$:$Mll2$ double conditional knockout myoblasts ($Mll1$:$Mll2$ dcKO), derived from $Rosa^{CE/+}$:$Mll1^{fl/fl}$, $Rosa^{CE/+}$:$Mll2^{fl/fl}$ and $Rosa^{CE/+}$:$Mll1^{fl/fl}$:$Mll2^{fl/fl}$ mice respectively. Myoblasts were treated for 7 days with 4-hydroxytamoxifen (4OHT) followed by a 2-day washout period (Supplementary Fig. 1a). The efficiency of $Mll1$ and $Mll2$ deletion was assessed at the RNA level (Fig. 1a–c), and the loss of MLL1 was further confirmed at the protein level (Supplementary Fig. 1b, c).

RT-qPCR analysis revealed that $Myf5$ mRNA was strikingly decreased in $Mll1$ cKO and $Mll1$:$Mll2$ dcKO myoblasts while its expression remained unchanged in $Mll2$ cKO myoblasts, indicating that MLL1, but not MLL2, regulates $Myf5$ expression in primary myoblasts (Fig. 1a–c). Interestingly, we also observed a mild but significant decrease of $Pax7$ mRNA level in both $Mll1$ cKO and $Mll1$:$Mll2$ dcKO myoblasts (Fig. 1a, c), indicating that MLL1 may also regulate $Pax7$ transcription. As we previously demonstrated that $Myf5$ is a target gene of the PAX7 transcription factor[11,13], our current results could suggest that $Myf5$ downregulation is due to a decrease in $Pax7$ expression.

To broadly examine changes in gene expression occurring after $Mll1$ deletion in primary myoblasts, we used microarray to compare RNA expression in control and $Mll1$ cKO myoblasts obtained after a 7-day culture period with either vehicle or 4-hydroxytamoxifen, respectively (Fig. 1d, Supplementary Fig. 1d). In $Mll1$ cKO myoblasts, 281 genes (322 probe sets) had a greater than twofold increase in expression, while 26 genes (29 probe sets), of which only 21 were for protein coding mRNAs, showed a greater than twofold decrease in gene expression (Table 1; Supplementary Data 1). Microarray confirmed that $Pax7$ expression was decreased in $Mll1$ cKO myoblasts (Table 1). Remarkably, among all the probe sets, $Myf5$ mRNA level displayed the greatest decrease (more than 10-fold). Initiation of myogenic differentiation coincides with both $Pax7$ and $Myf5$ downregulation, and $myogenin$ upregulation[19]. However, $myogenin$ mRNA level was unchanged in $Mll1$ cKO myoblasts (Fig. 1e; Table 1), ruling out the possibility that loss of MLL1 leads to precocious myogenic differentiation. Little to no change was observed in the expression of other genes involved in myogenesis, including $Myod1$ (Fig. 1d, e; Table 1). Interestingly, among the downregulated genes, 43% (9 out of 21) were direct MLL1 target genes in myoblasts[20], 4 of which were also Pax7 target genes[11]. RT-qPCR in $Mll1$ cKO myoblasts further confirmed the downregulation of several MLL1 target genes ($Asb4$, $Six2$, $Dlx1$ and $Lbx1$) as well as MLL1 and PAX7 common target genes ($Fgfr4$ and $Myf5$)[11,20,21] (Fig. 1e; Table 1).

Together, our data suggest that MLL1, but not MLL2, regulates $Pax7$ transcription in primary myoblasts. The reduced $Pax7$ expression in $Mll1$-deficient myoblasts could also explain the decreased expression of its target genes, $Myf5$ and $Fgfr4$.

**MLL1 binds both $Pax7$ and $Myf5$ genomic loci.** MLL1 possesses a specific methyltransferase activity responsible for the deposition of an active tri-methylation mark on the histone H3 lysine 4 (H3K4me3) allowing for the transcriptional activation of its target genes[22]. In primary myoblasts, MLL1 bound $Pax7$ and $Myf5$ loci near the transcriptional start site (Supplementary Fig. 2a). MLL1 also bound $Pax7$ +11.5 and +11.7 kb regions, which were previously shown to display H3K4me3 marks[23]. This result suggests that MLL1 can methylate the chromatin over both $Pax7$ and $Myf5$ promoters. To determine whether MLL1 binding corresponds to H3K4me3 enrichment at both $Myf5$ and $Pax7$ loci, we performed chromatin immunoprecipitation (ChIP)-qPCR using an anti-H3K4me3 antibody in control and $Mll1$ cKO myoblasts, and analyzed H3K4me3 enrichment at $Pax7$ and $Myf5$ promoters

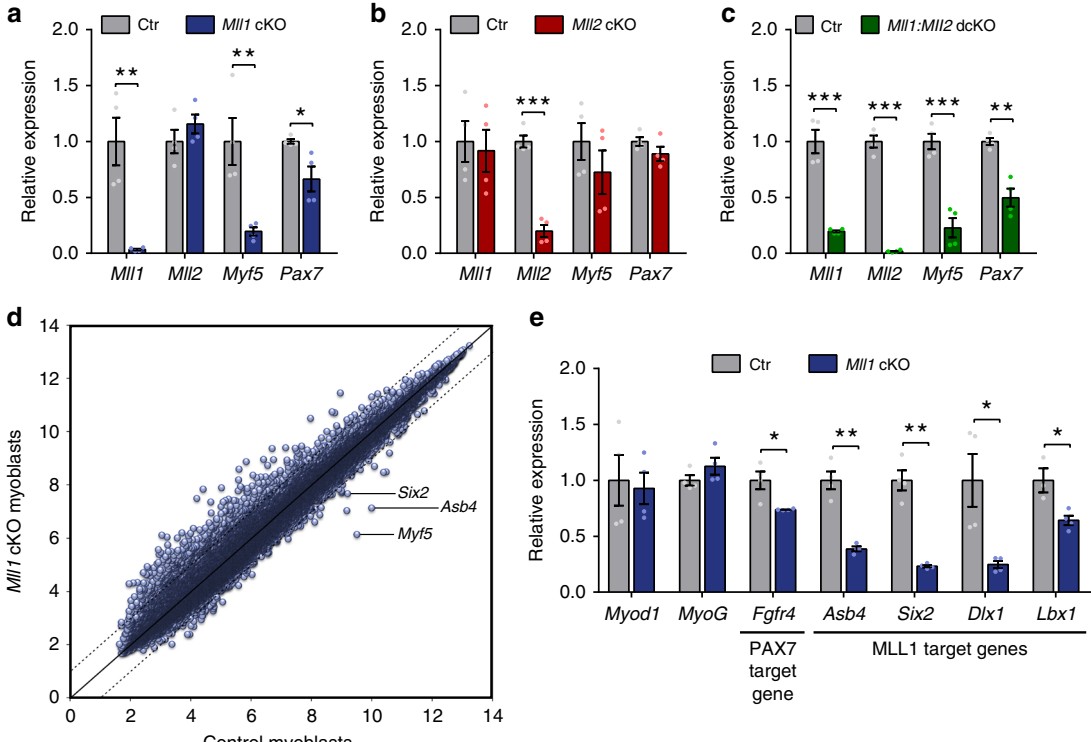

**Fig. 1** MLL1 is required for *Myf5* and *Pax7* expression in primary myoblasts. **a–c** Relative expression of *Mll1*, *Mll2*, *Myf5*, and *Pax7* in **a** *Mll1* cKO, **b** *Mll2* cKO, and **c** *Mll1:Mll2* dcKO primary myoblasts and their respective controls, as determined by RT-qPCR. Data are presented as the mean ± S.E.M. of four independent experiments. **d** Microarray analysis was performed on control and *Mll1* cKO primary myoblasts. A total of 35,557 probes were identified in the samples and plotted. **e** Relative expression of *Myod1*, *myogenin*, as well as PAX7 and MLL1 target genes in control and *Mll1* cKO primary myoblasts, as determined by RT-qPCR. Data are presented as the mean ± S.E.M. of four independent experiments

(Fig. 2a, b). *Mll1* deletion resulted in a clear and extensive decrease in H3K4me3 marks at the *Myf5* locus (Fig. 2c). A modest decrease in H3K4me3 enrichment was also observed at the *Pax7* locus (Fig. 2c), consistent with our RT-qPCR data showing a moderate decrease in *Pax7* mRNA level. More generally, we analyzed the total level of the histone H3 and the H3K4me3 mark between control and *Mll1* cKO myoblasts (Supplementary Fig. 2b–d). H3K4me3 immunoblot analysis did not reveal any differences (Supplementary Fig. 2d), suggesting that the loss of H3K4me3 following *Mll1* deletion is gene-specific rather than global in primary myoblasts. This was expected since there was only a small number of genes downregulated upon *Mll1* deletion (Table 1, Supplementary Data 1).

**Loss of MLL1 strongly impairs myoblast proliferation**. Our gene expression analysis revealed that *Mll1* deletion leads to a 40% decrease in *Pax7* mRNA expression. Interestingly, while 90% of the control myoblasts expressed PAX7 by immunostaining, only 30% of *Mll1* cKO myoblasts displayed a detectable signal for PAX7 (Fig. 3a, b; Supplementary Fig. 3a, b). Accordingly, western blot analysis revealed that PAX7 protein level was dramatically decreased in *Mll1* cKO myoblasts as compared with control (Fig. 3c, d). Since loss of the PAX7 transcription factor results in reduced proliferation[9], we assessed the proliferation of *Mll1* cKO myoblasts. After 6 days of culture, *Mll1* cKO myoblasts underwent a modest 3.5-fold expansion while control myoblasts expanded by 40-fold (Fig. 3e). To further confirm the impairment of proliferation following *Mll1* deletion, myoblasts were immunostained with the cell proliferation marker Ki67 (Fig. 3a). *Mll1* cKO myoblasts showed a twofold decrease in the proportion of Ki67-positive cells compared with the control (Fig. 3f).

We also evaluated the capacity of *Mll1* cKO myoblasts to differentiate. Either MYF5 or MYOD are capable of activating the myogenic program[24]. Although *Mll1* deletion resulted in loss of *Myf5* expression, *Mll1* cKO myoblasts still expressed MYOD as well as other myogenic markers such as SYNDECAN-4 (SDC4) (Fig. 3c; Table 1; Supplementary Fig. 3a, b). This result indicates that *Mll1* cKO myoblasts retained myogenic competence. *Mll1* cKO myoblasts plated at high confluency were able to differentiate, expressing both MYOGENIN and myosin heavy chain (MHC) (Fig. 3g; Supplementary Fig. 3c), and to fuse normally into myotubes (Fig. 3h). Nevertheless, after 6 days of differentiation, a pool of PAX7-expressing mononucleated myoblasts, or reserve cells, arose alongside the control myotubes whereas it was absent after *Mll1* deletion (Fig. 3h).

Taken together, this suggests that MLL1 is required to generate the reserve progeny upon differentiation. These results demonstrate that MLL1 is required for PAX7 expression and proliferation of myogenic progenitor cells, but is dispensable for differentiation.

**Exogenous *Pax7* rescues *Myf5* expression in *Mll1* cKO myoblasts**. *Mll1* cKO myoblasts display more than 10-fold lower levels of *Myf5* transcript than control myoblasts and scarcely express Pax7 protein (Figs. 1 and 3). The PAX7 transcription factor is known to recruit the Trithorax complex to *Myf5* regulatory sequences through a direct interaction with MLL1/2[13,14]. As PAX7 is required for *Myf5* expression[11,13], we hypothesized that the decrease of *Myf5* transcription in *Mll1* cKO primary myoblasts is directly attributed to the loss of PAX7 rather than the loss of MLL1. Hence, we asked whether the downregulation of *Myf5* expression in *Mll1* cKO myoblasts could be rescued by PAX7 overexpression solely.

**Table 1 Partial list of genes differentially expressed in *Mll1* cKO and control primary myoblasts**

| | Gene symbol | Array signal Control | Array signal *Mll1* cKO | Array Fold change | qPCR Fold change |
|---|---|---|---|---|---|
| | *Downregulated genes* | | | | |
| 1 | Myf5 | 9.51 | 6.14 | −10.33 | −20.00 |
| 2 | Asb4 | 9.99 | 7.12 | −7.31 | −14.28 |
| 3 | Six2 | 9.20 | 7.67 | −2.89 | −7.14 |
| 4 | Dlx1 | 9.00 | 7.74 | −2.40 | −2.85 |
| 5 | Fgfr4 | 9.00 | 7.77 | −2.34 | −2.43 |
| 6 | Olfr767 | 7.64 | 6.46 | −2.27 | nd |
| 7 | Lbx1 | 7.97 | 6.82 | −2.22 | −5.26 |
| 8 | Lmnb1 | 8.91 | 7.84 | −2.11 | nd |
| 9 | Cdc42ep3 | 8.69 | 7.64 | −2.07 | nd |
| 10 | Lbr | 9.04 | 8.00 | −2.05 | nd |
| | *Upregulated genes* | | | | |
| 1 | Slc40a1 | 5.84 | 8.87 | 8.16 | nd |
| 2 | Timp3 | 5.60 | 8.59 | 7.94 | nd |
| 3 | Stmn2 | 5.02 | 7.89 | 7.33 | nd |
| 4 | Dlk1 | 6.70 | 9.37 | 6.37 | 18.25 |
| 5 | Sfrp2 | 6.41 | 8.95 | 5.83 | 4.64 |
| 6 | Cd24a | 7.97 | 10.50 | 5.76 | 4.96 |
| 7 | Nnat | 7.15 | 9.40 | 4.75 | nd |
| 8 | RbEST47 | 6.75 | 8.95 | 4.62 | nd |
| 9 | Ly6a (Sca1) | 6.55 | 8.73 | 4.54 | 2.89 |
| 10 | Krt18 | 6.19 | 8.26 | 4.22 | nd |
| | *Genes related to myogenesis* | | | | |
| 1 | Myod1 | 12.15 | 12.27 | 1.08 | 1.33 |
| 2 | Myogenin | 7.30 | 7.94 | 1.56 | 1.17 |
| 3 | Myf6 | 5.40 | 5.07 | −1.26 | −1.70 |
| 4 | Mef2a | 11.08 | 11.12 | 1.03 | nd |
| 5 | Mef2b | 6.44 | 6.42 | −1.01 | nd |
| 6 | Mef2c | 6.25 | 7.69 | 2.71 | nd |
| 7 | Myh1 | 6.62 | 7.86 | 2.36 | nd |
| 8 | Pax3 | 6.44 | 6.99 | 1.47 | nd |
| 9 | Pax7 | 10.29 | 10.02 | −1.21 | −1.58 |
| | *MLL1 target genes* | | | | |
| 1 | Hoxa6 | 7.44 | 7.47 | 1.02 | nd |
| 2 | Hoxa7 | 7.64 | 7.88 | 1.18 | nd |
| 3 | Hoxa10 | 7.74 | 7.65 | −1.06 | nd |
| 4 | Hoxa11 | 8.28 | 7.81 | −1.38 | nd |
| 5 | Hoxc6 | 7.84 | 8.06 | 1.16 | nd |
| 6 | Hoxc8 | 8.03 | 7.59 | −1.36 | nd |
| 7 | Hoxc9 | 8.44 | 8.69 | 1.19 | nd |
| 8 | Hoxc10 | 8.51 | 8.65 | 1.10 | nd |
| 9 | Dlx2 | 10.44 | 10.79 | 1.27 | nd |
| 10 | Prdm16 | 6.41 | 5.85 | −1.47 | nd |
| | *Histone methyltransferases* | | | | |
| 1 | Mll1 (Kmt2a) | 9.44 | 9.66 | 1.17 | nd |
| 2 | Mll2 (Kmt2b) | 7.90 | 8.05 | 1.11 | nd |
| 3 | Setd1a | 8.81 | 8.88 | 1.05 | nd |
| 4 | Setd1b | 8.52 | 8.42 | −1.07 | nd |

Microarray analysis was performed in myoblasts isolated from control and *Mll1* cKO myoblasts. Change in expression for a subset of genes was further confirmed by qPCR as indicated

To address this question, we generated stable $Rosa^{CE/+}:Mll1^{fl/fl}$ cell lines overexpressing either an empty plasmid (CMV-empty) or *Pax7* (CMV-Pax7), and we subsequently treated these lines with either vehicle (control) or 4-hydroxytamoxifen (*Mll1* cKO) (Supplementary Fig. 4a). *Mll1* deletion was confirmed at both RNA and protein levels (Supplementary Fig. 4b–d). As expected, upon 4-hydroxytamoxifen treatment, $Rosa^{CE/+}:Mll1^{fl/fl}$ cells expressing the empty plasmid exhibited a 40% decrease and a 90% decrease in *Pax7* and *Myf5* transcripts, respectively (Fig. 4b, c). Consistent with our previous finding, PAX7 protein was barely detectable (Fig. 4a; Supplementary Fig. 4c). Compared with the control lines, CMV-Pax7 transduced myoblasts had increased and stable PAX7 expression at both RNA and protein levels independently of *Mll1* deletion (Fig. 4a, b; Supplementary Fig. 4c).

*Pax7*-overexpressing myoblasts also exhibited increased *Myf5* expression compared with the myoblasts transduced with the empty vector (Fig. 4c). Importantly, deletion of *Mll1* in *Pax7*-overexpressing myoblasts neither changed *Myf5* expression, nor H3K4me3 enrichment at the *Myf5* locus (Fig. 4c, d), demonstrating that the loss of *Myf5* expression and H3K4me3 at the *Myf5* locus in *Mll1* cKO myoblasts is a consequence of the absence of PAX7. Of note, *Pax7* overexpression in *Mll1* cKO myoblasts did not rescue the expression of the MLL1 target genes *Six2*, *Dlx1*, *Lbx1*, or *Asb4*, as their expression was decreased in *Pax7*-overexpressing cells following *Mll1* deletion (Fig. 4e). However, this phenotype was not observed for the PAX7 target gene *Fgfr4*[11] (Fig. 4e). This indicates that *Mll1* deletion results in changes in gene expression that can be partially rescued by PAX7, and

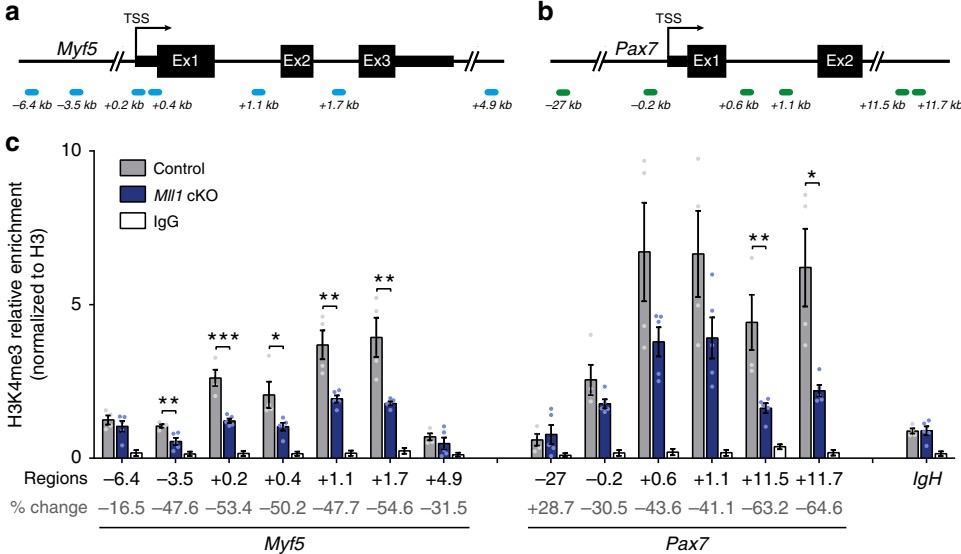

**Fig. 2** Loss of MLL1 decreases H3K4me3 marks at *Myf5* and *Pax7* loci in *Mll1* cKO myoblasts. **a** Location of primer sets relative to *Myf5* transcription start site (TSS) for ChIP-qPCR analysis of *Myf5* promoter. **b** Location of primer sets relative to *Pax7* TSS and its downstream region for ChIP-qPCR analysis of *Pax7* promoter. **c** ChIP-qPCR analysis of H3K4me3 enrichment, normalized to total H3, at *Myf5* and *Pax7* loci in control and *Mll1* cKO myoblasts, obtained after 4OHT-treated or EtOH-treated *Rosa^CE/+:Mll1^fl/fl* myoblasts ($n = 5$ independent experiments). *IgH* enhancer region is used as a negative control. Percent change in H3K4me3 enrichment between control and *Mll1* cKO myoblasts is indicated for each region

suggests that other histone methyltransferases can compensate for the absence of MLL1 to activate PAX7 target genes in *Mll1* cKO myoblasts.

In order to evaluate whether another histone methyltransferase (HMT) can substitute for MLL1 in regulating the expression of PAX7 and its target genes, we transiently transfected primary myoblasts with siRNAs targeting *Mll1* alone or in combination with siRNAs targeting *Mll2*, *Set1a*, and *Set1b*, which are known to mediate H3K4 tri-methylation at gene promoters[15,16] (Supplementary Fig. 4e, f). As a control, we first tested the capacity of the four individual HMTs to regulate *Pax7* and *Myf5* transcription (Supplementary Fig. 4e). We observed that only *Mll1* knockdown leads to a decrease in both *Pax7* and *Myf5* expression (Supplementary Fig. 4e), recapitulating our first observations (Fig. 1). Interestingly, co-transfection of *Mll1* siRNA with either *Mll2* siRNA or *Set1b* siRNA further decreased *Pax7* and *Myf5* expression, while the combination of *Mll1* and *Set1a* siRNA did not show any effect compared with the treatment with *Mll1* siRNA only (Supplementary Fig. 4f). Thus, these results suggest that in *Mll1* cKO myoblasts, MLL2 and SET1B can compensate and substitute for MLL1 in regulating *Pax7* and *Myf5* RNA levels.

Consistent with the finding that PAX7 regulates myoblast proliferation[9], *Pax7*-overexpressing myoblasts exhibited increased proliferation compared with the myoblasts transduced with the empty vector (Fig. 4f). However, the proliferation defect observed in *Mll1* cKO myoblasts was only partially rescued by PAX7 overexpression as demonstrated by Ki67 immunostaining and proliferation assay (Fig. 4a, f). Taken together, our data confirm that MLL1 is required for PAX7 expression and myoblast proliferation, and that PAX7 expression is sufficient to restore the transcription of its target genes. In contrast, the proliferation defect in *Mll1* cKO myoblasts is partially rescued by PAX7 suggesting that MLL1 can mediate cell proliferation independently of PAX7.

**Mll1 deletion in satellite cells impairs muscle regeneration.** PAX7 is required for satellite cell function in adult skeletal muscle[5,7,9,10]. Since PAX7 expression is compromised in *Mll1*

cKO myoblasts, we investigated the role of MLL1 in satellite cell function in vivo. To this end, we generated *Pax7^CE/+:Mll1^fl/fl*: *Rosa^YFP* mice allowing for tamoxifen-inducible genetic deletion of *Mll1* in *Pax7*-expressing cells (Supplementary Fig. 5a).

*Pax7^CE/+:Mll1^fl/fl:Rosa^YFP* mice were intraperitoneally injected with tamoxifen for 4 consecutive days and kept on a tamoxifen diet thereafter (Fig. 5a). Since our in vitro data revealed that MLL1 is required for both PAX7 expression and myoblast proliferation, *Pax7^CE/+:Mll1^fl/fl:Rosa^YFP* mice were maintained on tamoxifen diet throughout the experimental time course to prevent growth of satellite cells escaping from tamoxifen-induced *Mll1* recombination[9]. Non-tamoxifen-treated *Pax7^CE/+:Mll1^fl/fl*: *Rosa^YFP* mice were used as controls. One week after the first tamoxifen injection, the efficiency of *Mll1* deletion was determined by RT-qPCR on freshly isolated quiescent satellite cells and activated satellite cells sorted 3 days after cardiotoxin (CTX)-induced injury (Supplementary Fig. 5b, d). Although *Mll1* was significantly downregulated in quiescent satellite cells after tamoxifen treatment, we did not observe differences in the expression of *Pax7* and its target genes *Myf5* and *Fgfr4*, while the expression of MLL1 target genes were downregulated (Supplementary Fig. 5b). Moreover, H3K4me3 enrichment was unchanged at both *Pax7* and *Myf5* loci (Supplementary Fig. 5c). Together, these results suggest that MLL1 does not regulate PAX7 expression in quiescent satellite cells.

However, as observed for the *Mll1* cKO primary myoblasts, RT-qPCR revealed that the expression of *Pax7* and its target genes *Myf5* and *Fgfr4* was significantly downregulated in *Mll1*-deficient activated satellite cells (Supplementary Fig. 5d). MLL1 target genes were also downregulated. Consistent with the RT-qPCR data, H3K4me3 enrichment was decreased at both *Pax7* and *Myf5* loci in activated satellite cells (Supplementary Fig. 5e). Together, these data indicate that MLL1 is important in regulating Pax7 expression upon satellite cell activation.

We also addressed whether *Mll1* deletion impairs satellite cell function by studying the capacity of *Mll1* cKO satellite cells to regenerate the muscle after an injury. Muscle regeneration was assessed at 7 days and 21 days following acute injury induced by cardiotoxin (CTX) injection into the *tibialis anterior* (TA) muscle

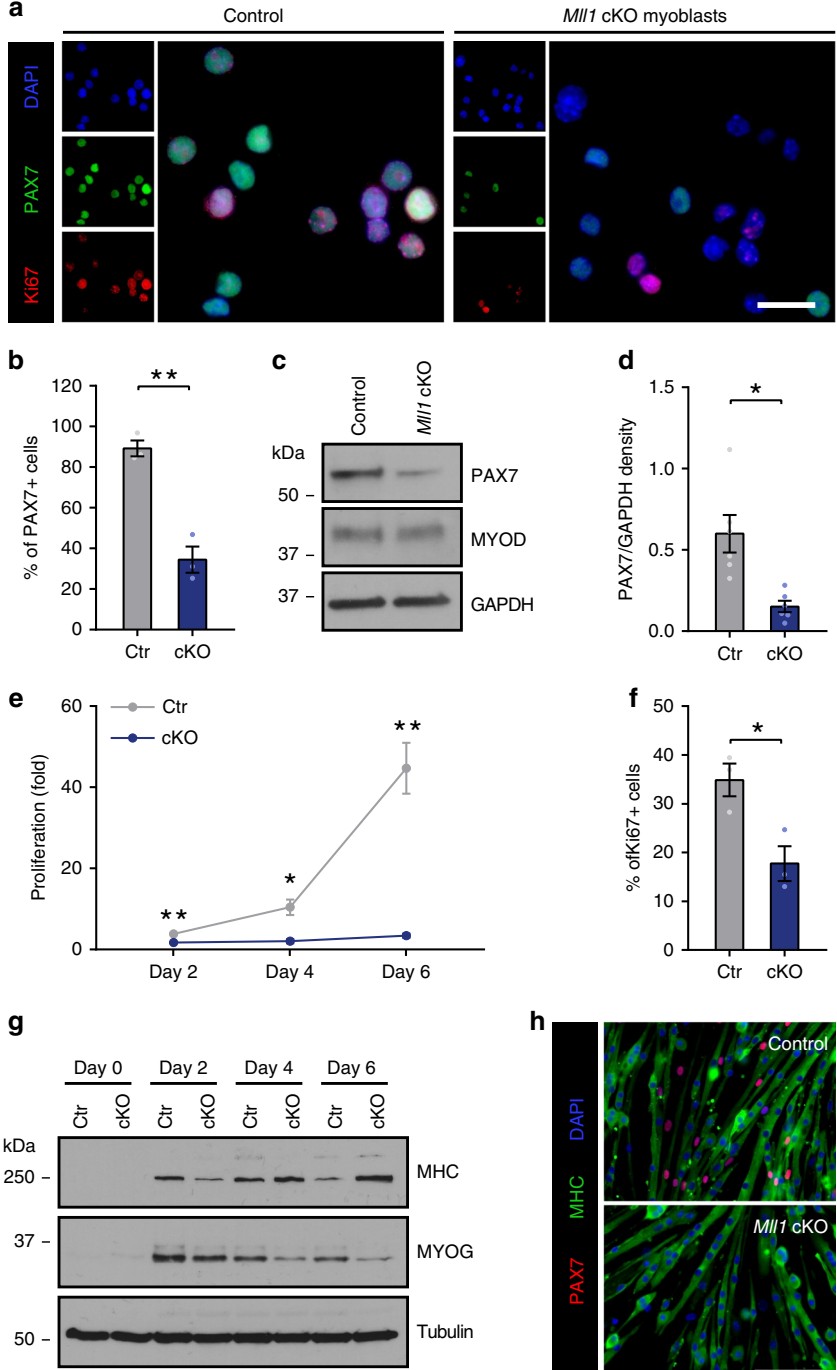

**Fig. 3** Loss of MLL1 abolishes PAX7 expression and myoblast proliferation, but does not affect myogenic differentiation. **a** Immunostaining of PAX7 (green) and Ki67 (red) in control and *Mll1* cKO primary myoblasts. Nuclei are counterstained with DAPI (blue). Scale bars represent 20 μm. **b** Quantification of PAX7-expressing cells from control and *Mll1* cKO primary myoblasts. Data are presented as the mean ± S.E.M. of three independent experiments. **c** Western blot analysis of PAX7 and MYOD in control and *Mll1* cKO primary myoblasts. GAPDH is used as a loading control. **d** Densitometric analysis of the relative levels of PAX7 normalized to GAPDH signals of six independent experiments. Values are represented as the mean ± S.E.M. **e** Control and *Mll1* cKO primary myoblasts were plated at the same density and grown for 6 days. Every 2 days, cells were numerated to evaluate their proliferation, represented as the mean (n = 3 biological replicates) ± S.E.M. **f** Quantification of Ki67-expressing cells from control and *Mll1* cKO primary myoblasts. Data are presented as the mean (n = 3 biological replicates) ± S.E.M. **g** Western blot analysis of Myosin Heavy Chains (MHC) and MYOGENIN during differentiation of control and *Mll1* cKO primary myoblasts. TUBULIN is used as a loading control. **h** Immunostaining of MHC (green) and PAX7 (red) in control and *Mll1* cKO primary myoblasts cultured in differentiation medium for 6 days. Nuclei are counterstained with DAPI (blue). Scale bar represents 100 μm

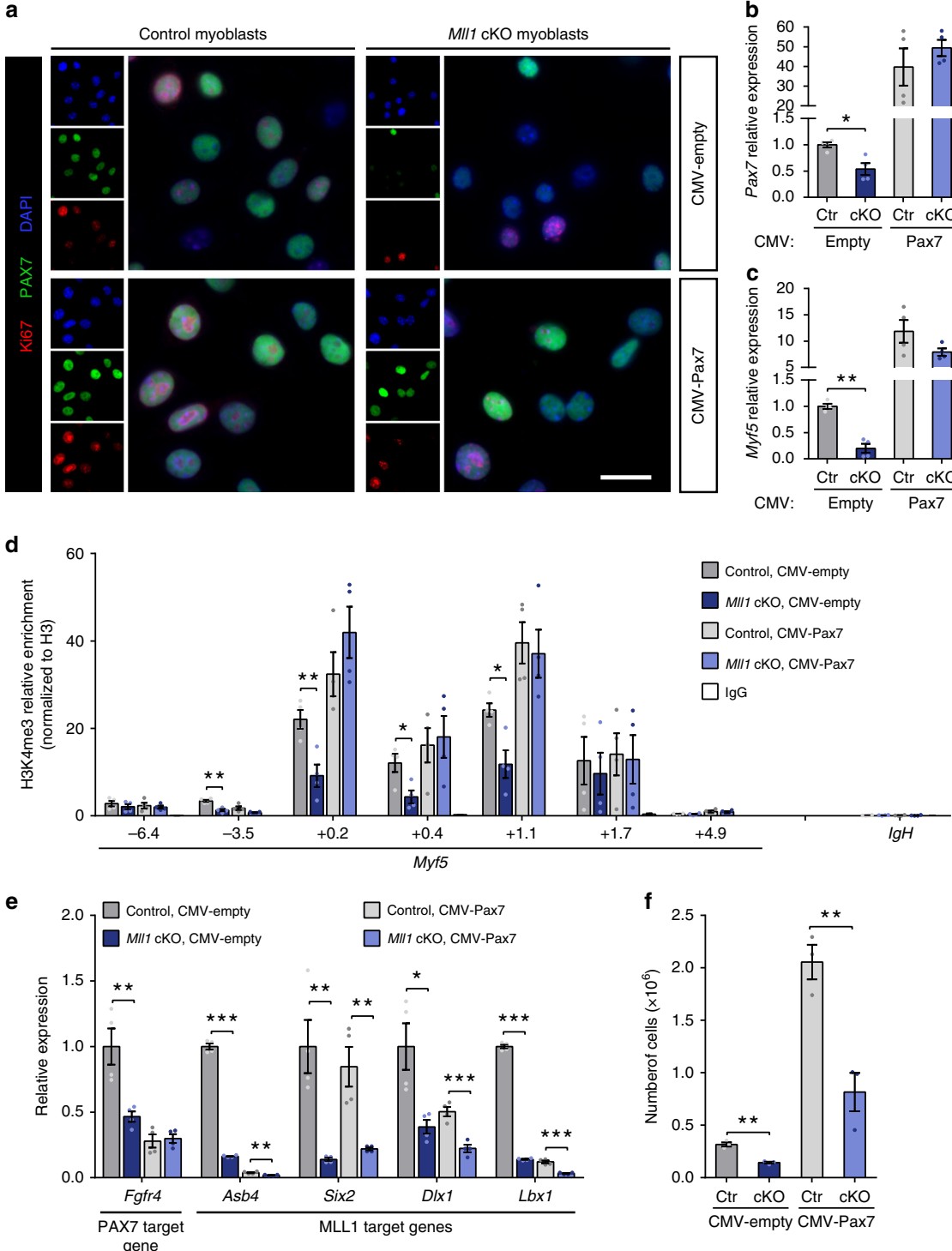

**Fig. 4** Overexpression of PAX7 restores *Myf5* expression and H3K4me3 marks at *Myf5* locus in *Mll1* cKO myoblasts. **a** Immunostaining of PAX7 (green) and Ki67 (red) in control and *Mll1* cKO primary myoblasts transduced with either an empty plasmid or a CMV-Pax7 plasmid. Nuclei are counterstained with DAPI (blue). Scale bars represent 20 μm. **b**, **c** RT-qPCR analysis of (**b**) *Pax7* and (**c**) *Myf5* mRNA expression performed on control and *Mll1* cKO primary myoblasts transduced with either an empty plasmid or a CMV-Pax7 plasmid, represented as mean ± S.E.M. ($n = 4$ independent experiments). **d** ChIP-qPCR analysis of H3K4me3 enrichment, normalized to H3, at *Myf5* locus in control and *Mll1* cKO primary myoblasts transduced with either an empty plasmid or a CMV-Pax7 plasmid. Values represent mean ± S.E.M. ($n = 3$ independent experiments). *IgH* enhancer region is used as a negative control. **e** Relative expression of PAX7 and MLL1 target genes in control and *Mll1* cKO primary myoblasts transduced with either an empty plasmid or a CMV-Pax7 plasmid, represented as mean ± S.E.M. ($n = 4$ independent experiments). **f** Number of control and *Mll1* cKO primary myoblasts transduced with either an empty plasmid or a CMV-Pax7 plasmid. Cells were plated at the same density and grown for 6 days before enumeration. Values represent mean ± S.E.M. ($n = 4$ independent experiments)

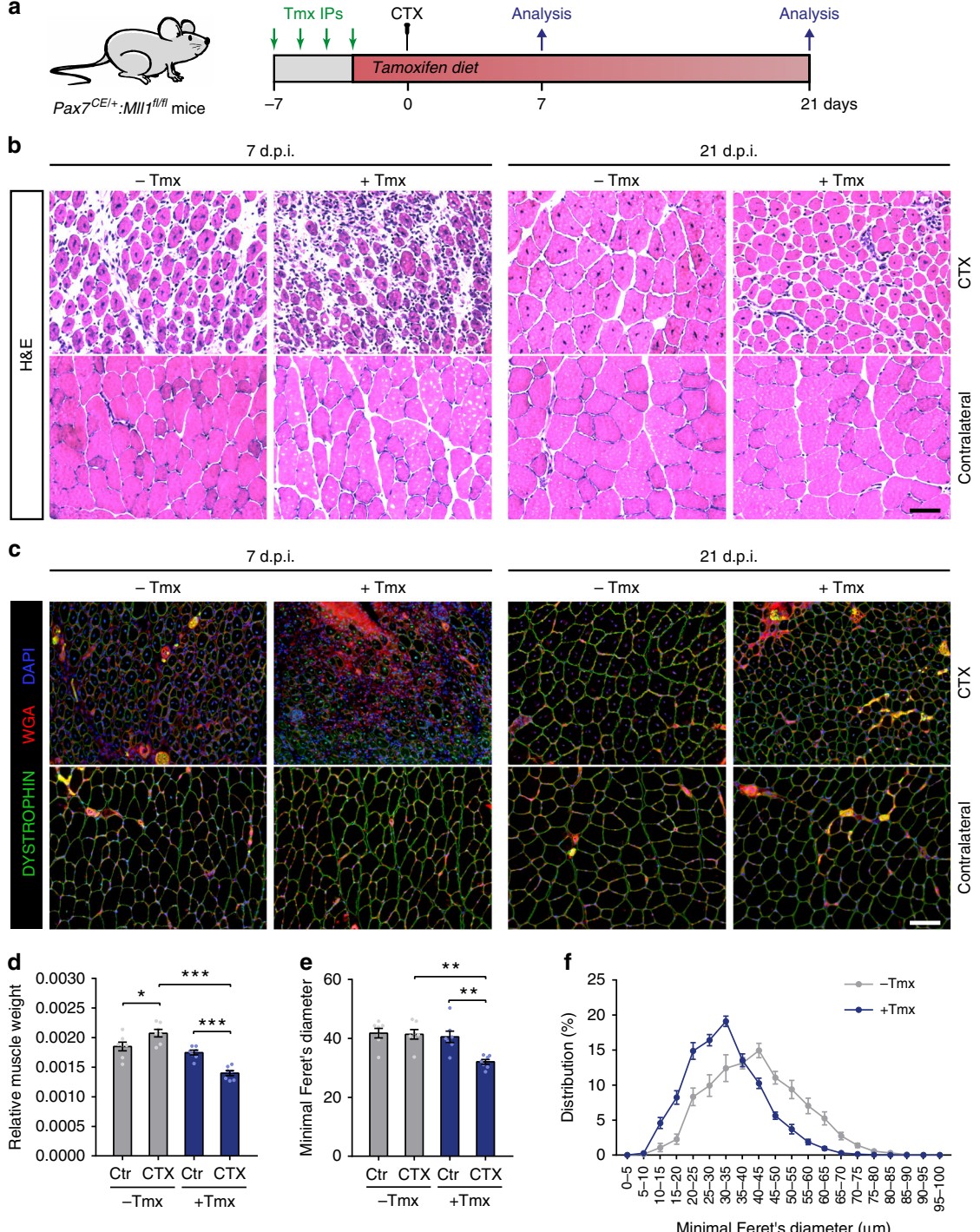

**Fig. 5** Genetic deletion of *Mll1* in satellite cells strongly impairs muscle regeneration. **a** Experimental schematic outlining the protocol followed to analyzed muscle regeneration of *Pax7^(CE/+)*:Mll1^(fl/fl)* mice. **b** Hematoxylin and eosin (H&E) staining of cardiotoxin (CTX)-injured and contralateral (ctr) *tibialis anterior* (TA) muscles from *Pax7^(CE/+)*:Mll1^(fl/fl)* mice, treated or not with tamoxifen at 7 days or 21 days post-injury. Scale bar represents 100 μm. **c** Wheat germ agglutinin (WGA, red) and DYSTROPHIN (green) staining of CTX-injured and contralateral TA muscles from *Pax7^(CE/+)*:Mll1^(fl/fl)* mice, treated or not with tamoxifen at 7 days or 21 days post-injury. Nuclei are stained with DAPI (blue). Scale bar represents 100 μm. **d** Relative CTX-injured and contralateral (ctr) TA muscle weight normalized to the body weight at 21 days post-injury, represented as means ± S.E.M (n = 7 mice per group). **e** Myofiber minimal Feret's diameter of CTX-injured and contralateral (ctr) TA muscles from *Pax7^(CE/+)*:Mll1^(fl/fl)* mice treated or not with tamoxifen at 21 days post-injury, represented as mean ± S.E.M (n = 7 mice per group). **f** Distribution of minimal Feret's diameter of CTX-injured TA muscles, as determined in **e**

(Fig. 5a). Hematoxylin and eosin (H&E) and wheat germ agglutinin (WGA) staining on muscle cross-sections revealed evident histological differences in tamoxifen-treated *Pax7^(CE/+)*: Mll1^(fl/fl)*:Rosa^(YFP)* mice associated with increased fibrotic deposits

at 7 days post-injury (d.p.i.) and smaller myofibers at 21 d.p.i. (Fig. 5b, c). Tamoxifen-treated *Pax7^(CE/+)*:Mll1^(fl/fl)*:Rosa^(YFP)* mice exhibited a 30% decrease in injured muscle weight compared with the control mice at 21 d.p.i. (Fig. 5d; Supplementary Fig. 5f).

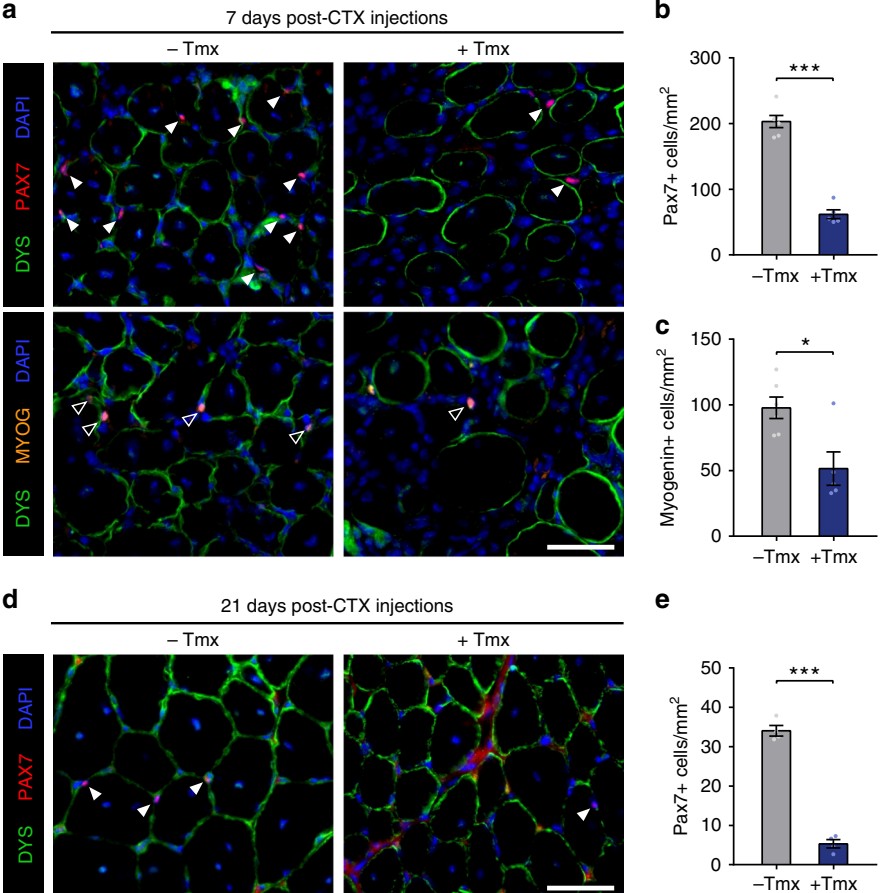

**Fig. 6** MLL1 is required for satellite cell proliferation and maintenance of the satellite cell pool. **a** Immunostaining with PAX7 (red) and MYOGENIN (orange) showing a markedly reduced numbers of satellite cells (white arrows) and differentiated cells (empty arrows) after satellite cell-specific deletion of *Mll1* induced by tamoxifen treatment. DYSTROPHIN staining (green) delineates the myofibers. Nuclei are counterstained with DAPI (blue). Scale bar represents 50 μm. **b** Quantification of Pax7-expressing cells per mm² from injured TA muscles of untreated (*n* = 6 mice) or tamoxifen-treated (*n* = 5 mice) *Pax7^CE/+:Mll1^fl/fl* mice at 7 days post-injury. **c** Quantification of MYOGENIN-expressing cells per mm² from injured TA muscle sections of untreated (*n* = 6 mice) or tamoxifen-treated (*n* = 5 mice) *Pax7^CE/+:Mll1^fl/fl* mice 7 days post-injury. **d** Immunostaining with PAX7 (red) confirming the decreased number of satellite cells (white arrows) after satellite cell-specific deletion of *Mll1* induced by tamoxifen treatment. DYSTROPHIN staining (green) delineates the myofibers. Nuclei are counterstained with DAPI (blue). Scale bar represents 50 μm. **e** Quantification of Pax7-expressing cells per mm² from injured TA muscle sections of untreated or tamoxifen-treated *Pax7^CE/+:Mll1^fl/fl* mice 21 days post-injury (*n* = 4 mice per group)

While the total number of myofibers in injured muscles was unchanged (Supplementary Fig. 5g), myofiber minimal Feret's diameter was significantly decreased in tamoxifen-treated mice at 21 d.p.i. (Fig. 5e, f). No differences were observed in the uninjured contralateral muscles, consistent with the hypothesis that MLL1 does not regulate PAX7 in the quiescent stem cell pool. Taken together, our results indicate that loss of MLL1 in satellite cells strongly impairs muscle regeneration.

**MLL1 is required for satellite cell self-renewal.** The impairment in muscle regeneration implies defects in satellite cell function of tamoxifen-treated *Pax7^CE/+:Mll1^fl/fl:Rosa^YFP* mice. Therefore, we analyzed the functional consequences of *Mll1* deletion in satellite cells. At 7 d.p.i., we enumerated the numbers of proliferating and differentiating satellite cells expressing either PAX7 or MYO-GENIN (MYOG), respectively (Fig. 6a). Enumeration of PAX7-positive cells revealed a dramatic fourfold decrease in satellite cell numbers (Fig. 6b), consistent with reduced cell proliferation. At 7 d.p.i., tamoxifen-treated *Pax7^CE/+:Mll1^fl/fl:Rosa^YFP* mice also exhibited a twofold decrease in the numbers of MYOGENIN-expressing cells (Fig. 6c). The loss of MYOG-expressing cells could be a consequence of the reduced number of satellite cells.

Alternatively, it could suggest a differentiation defect of *Mll1*-deficient satellite cells. At the end of the regeneration process at 21 d.p.i., the number of PAX7-expressing satellite cells was reduced by sixfold in tamoxifen-treated *Pax7^CE/+:Mll1^fl/fl:Rosa^YFP* mice (Fig. 6d, e), suggesting that *Mll1* knockout satellite cells failed to self-renew and *Mll1* deletion led to the loss of the satellite stem cell pool.

To examine the fate of satellite cells in tamoxifen-treated *Pax7^CE/+:Mll1^fl/fl:Rosa^YFP* mice during muscle regeneration, we took advantage of the *Rosa^YFP* allele to label PAX7-expressing *Mll1*-deficient satellite cells and trace them during muscle regeneration (Supplementary Fig. 5a). One week after the first tamoxifen injection, immunostaining on muscle cross-sections revealed that YFP was expressed specifically in PAX7-expressing satellite cells in tamoxifen-treated *Pax7^CE/+:Mll1^fl/fl:Rosa^YFP* mice (Fig. 7a; Supplementary Fig. 6a). No YFP expression was detected in non-tamoxifen-treated mice. At 7 d.p.i., YFP-positive cells adjacent to the regenerating myofibers were detected, all of them expressing either PAX7 or MYOG (Fig. 7b). Interestingly, YFP was also expressed in the majority of regenerating myofibers marked by the expression of the embryonic MHC (eMHC) (Fig. 7b; Supplementary Fig. 6b), suggesting that YFP-positive satellite cells were able to differentiate. At 21 d.p.i., the regenerated

  9

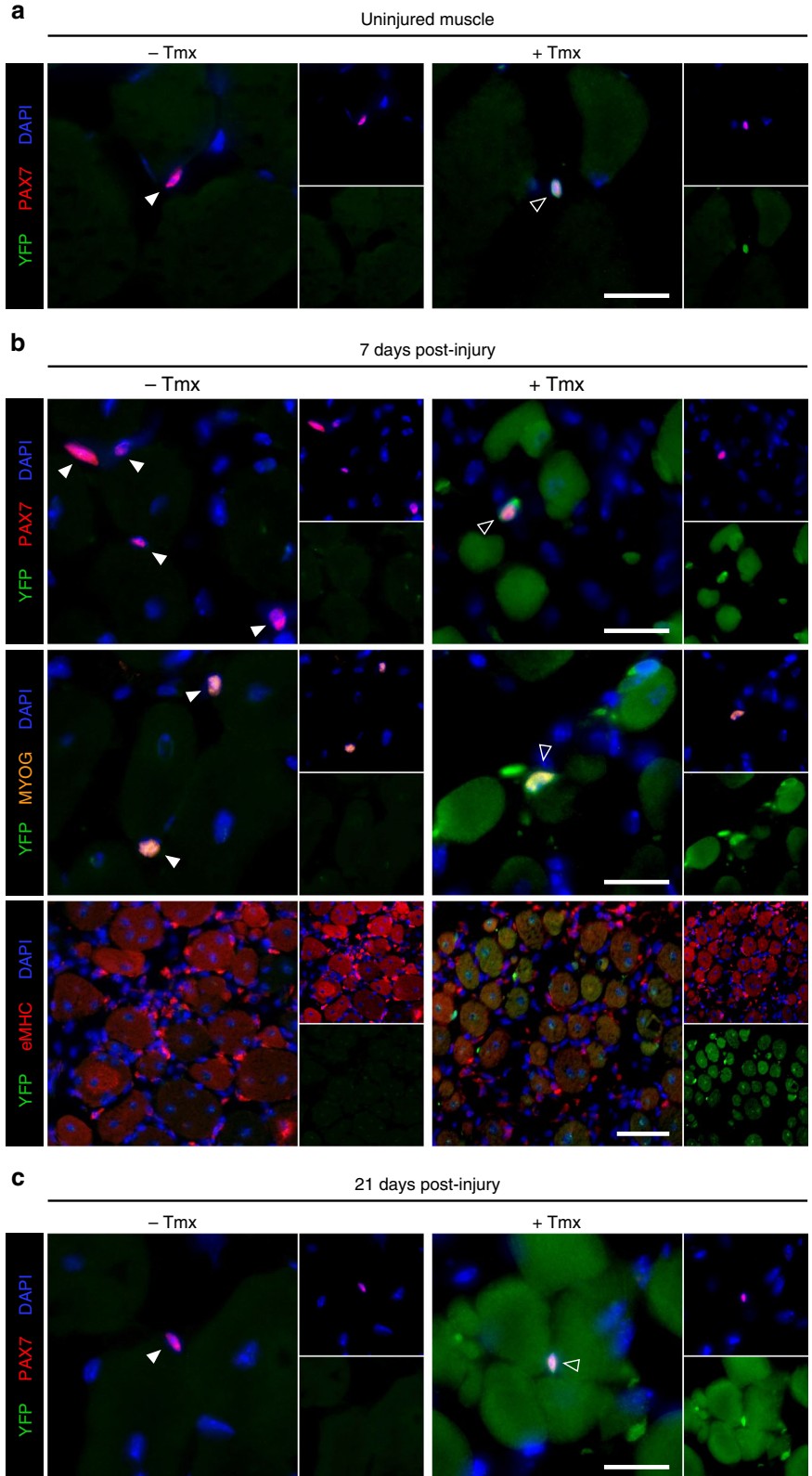

myofibers were all labeled by YFP, while only very rare YFP-positive PAX7-positive satellite cells were observed (Fig. 7c; Supplementary Fig. 6c). No interstitial YFP-positive cells were found. These observations demonstrate that *Mll1*-deficient satellite cells successfully differentiated and fused into new myofibers while they were not able to self-renew. Interestingly, the number of PAX7-positive satellite cells co-expressing YFP in

the contralateral (uninjured) muscle was unchanged from day 0 to day 21 post-injury (Supplementary Fig. 6) confirming that *Mll1* deficiency had no effect on PAX7 expression in the quiescent satellite cell pool.

To further confirm the loss of the satellite stem cell pool after regeneration, *Pax7^{CE/+}:Mll1^{fl/fl}:Rosa^{YFP}* mice were subjected to a second CTX injury (Supplementary Fig. 7a). Twenty one days

**Fig. 7** *Mll1*-deficient satellite cells retain their capacity to differentiate during muscle regeneration. **a** Immunostaining of PAX7 (red) showing satellite cells co-expressing YFP (green) upon 7-day tamoxifen treatment (+Tmx) in uninjured TA muscles from *Pax7*<sup>CE/+</sup>:*Mll1*<sup>fl/fl</sup>:*Rosa*<sup>YFP</sup> mice. Untreated mice (− Tmx) do not exhibit any YFP staining. Nuclei are stained with DAPI (blue). Scale bar represents 20 μm. **b** Top and mid panel: satellite cells expressing either PAX7 (red) or MYOGENIN (orange), co-stained with YFP (green), of CTX-injured TA muscles from *Pax7*<sup>CE/+</sup>:*Mll1*<sup>fl/fl</sup>:*Rosa*<sup>YFP</sup> mice treated or not with tamoxifen at 7 days post-injury. Nuclei are counterstained with DAPI (blue). Scale bar represents 20 μm. Lower panel: immunostaining of embryonic MHC (eMHC, red) for regenerating myofibers and YFP (green) at 7 days post-injury. Scale bar represents 50 μm. **c** PAX7 (red) and YFP (green) immunostaining of CTX-injured TA muscles from *Pax7*<sup>CE/+</sup>:*Mll1*<sup>fl/fl</sup>:*Rosa*<sup>YFP</sup> mice treated or not with tamoxifen at 21 days post-injury. DAPI stained the nuclei (blue). Scale bar represents 20 μm. Throughout, white arrows and empty arrows indicate wild-type and *Mll1* knockout satellite cells, respectively

after the second injury, tamoxifen-treated $Pax7^{CE/+}:Mll1^{fl/fl}:Rosa^{YFP}$ mice exhibited a dramatic 70% decrease in the relative muscle weight compared with the controls (Supplementary Fig. 7b, c). The regeneration deficit was even more pronounced as evidenced by extensive fibrosis and absence of newly regenerated myofibers (Supplementary Fig. 7d–f). Finally, no PAX7-expressing cells were found in double injured muscles while satellite cells remained present in the contralateral muscle and in animals not treated with tamoxifen (Supplementary Fig. 7g). Therefore, these data clearly demonstrate that *Mll1* deletion leads to the loss of the satellite stem cell pool after regeneration.

Based on all our results, we conclude that MLL1 does not affect the quiescent stem cell pool in resting muscle. However, MLL1 is absolutely required for satellite cell proliferation and self-renewal during skeletal muscle regeneration, while it does not interfere with their capacity to differentiate.

## Discussion

Satellite cell function relies on the transcription factor PAX7, as it regulates a large number of target genes involved in myogenic identity, growth, and proliferation[5,9,11]. In this study, we show that MLL1, but not MLL2, is absolutely required for PAX7 expression in activated satellite cells and myoblasts, but not in quiescent satellite cells. PAX7 is capable of interacting with either MLL1 or MLL2[13,14]. However, only *Mll1* deletion leads to a complete loss of *Myf5* mRNA, while *Mll2* cKO myoblasts maintain *Myf5* transcription. These results indicate that in primary myoblasts, MLL2 is dispensable for *Myf5* expression. Accordingly, accumulating evidence shows that the activities of MLL1 and MLL2 are restricted to selective sets of target genes in very specific cell types during development[25–27]. Thus, MLL2 could be considered having a role in *Myf5* transcription through its interaction with PAX7 earlier during muscle development or on a distinct subset of target genes.

Aside from *Myf5* downregulation, our microarray analysis performed on control and *Mll1* cKO myoblasts revealed remarkably little change in gene expression, even amongst *Hox* and *homeobox* genes which have been previously determined to be MLL1 targets[20–22,28–31]. Cultured myoblasts are committed and have limited cell-type plasticity under standard growth conditions[2]. Thus, *Mll1* deletion would not result in a change in cell identity, explaining why we do not observe major modifications in *Hox* and *homeobox* gene expression. Nevertheless, we determined that 43% of the downregulated genes in *Mll1* cKO myoblasts were direct MLL1 target genes[20]. In contrast, upregulation of gene expression results mostly from an indirect effect of the loss of MLL1, since only 23% (64 of 281) of the upregulated genes were bound by MLL1 in myoblasts[20]. Gene ontology analysis on the upregulated genes revealed biological process (BP) terms associated with negative regulation of biological and cellular processes such as cell proliferation or migration, as expected (Supplementary Data 2). It also highlighted cellular component (CC) terms associated to the myofibril, contractile fiber, and sarcomere, further confirming that *Mll1* cKO cells retain myogenic identity. Our results indicate that in primary myoblasts, the

absence of MLL1 leads to the downregulation of a very small subset of genes including five transcription factors (*Pax7*, *Myf5*, *Lbx1*, *Six2*, and *Dlx1*), which in turn leads to an overall indirect increase in gene expression.

Overexpression of PAX7 rescues *Myf5* transcripts levels in the absence of MLL1, supporting the idea that *Myf5* expression is dependent on PAX7[11,13]. Moreover, PAX7 overexpression re-establishes the H3K4me3 marks on *Myf5* regulatory sequences in the absence of MLL1, suggesting that some other histone methyltransferases can substitute for MLL1 in activating the *Myf5* locus. Indeed, our results suggest that MLL2 and SET1B can compensate for the absence of MLL1 in regulating *Myf5* expression. Several studies have highlighted the robust activity of SET1A/B methyltransferase containing complexes, which write the majority of H3K4me3 marks relative to the MLL1–4 containing complexes[32–34]. Therefore, SET1B may be recruited to the *Myf5* promoter to activate transcription in the absence of MLL1.

The mechanism by which MLL1 is recruited to the *Pax7* promoter is still unknown. Chromatin immunoprecipitation experiments showed MLL1 binding to five genomic loci of the *Pax7* gene (−0.2 kb, +0.6 kb, +1.1 kb, +11.5 kb, +11.7 kb). In silico analysis revealed putative binding sites for several transcription factors that are expressed in satellite cells and bind at least two different *Pax7* loci, including KLF5, EWSR1, SP2, SP3, and ZFP263. Of note, an interaction between EWSR1 and MLL1 has been reported using BioID[35], and different oncogenic fusion proteins involving EWSR1 can drive *Pax7* expression in Ewing sarcomas[36]. Whether EWSR1 or other transcription factors are responsible for MLL1 recruitment to the *Pax7* promoter remains to be determined. Although MLL1 binds *Pax7* regulatory sequences, *Mll1* deletion in primary myoblasts only results in a modest decrease of H3K4me3 at the *Pax7* locus, in agreement with previous findings stating that loss of MLL1 does not always trigger H3K4me3 changes at its target genes[25,28,37]. In *Mll1* cKO myoblasts, *Pax7* mRNA levels are decreased by 40% compared with control, while PAX7 protein is barely detectable. Taken together, these results suggest the intriguing possibility that MLL1 contributes to PAX7 regulation through other mechanisms than its histone methyltransferase function. This hypothesis is further supported by several studies suggesting that the role of MLL1 is partly independent of its H3K4 methyltransferase activity[25,37,38]. Further studies will be required to clarify the non-HMT roles of MLL1 in muscle stem cells.

In accordance with the absolute requirement for PAX7-expressing satellite cells in skeletal muscle regeneration[5,9,39,40], satellite cell-specific deletion of *Mll1* dramatically impaired muscle regeneration in vivo. To date, MLL1 is the first known regulator of *Pax7* to display such a drastic phenotype that results in the failure in muscle regeneration and the exhaustion of the satellite stem cell pool, phenocopying the *Pax7*-conditional knockout mice[9]. This appears to be largely due to a defect in satellite cell proliferation and self-renewal. Indeed, during skeletal muscle regeneration, *Mll1*-deficient satellite cells fail to proliferate, but do ultimately differentiate and fuse into regenerating myofibers. At 21 days post-injury, only very rare

PAX7-expressing cells were observed, which correlates with a loss of the satellite stem cell pool. This was further supported by our results following a second muscle injury that ultimately results in a profound regeneration deficit as observed in different mouse models following genetic ablation of adult muscle stem cells[39,40]. However, all regenerating myofibers were YFP-positive, confirming that satellite cells have differentiated at the expense of self-renewal. Together, these results confirm that MLL1 plays essential roles in the proliferation and self-renewal of satellite cells.

Our data further support prior findings that MLL1 is a necessary factor for stem cell maintenance and function, and may suggest that MLL1 has important roles for other stem cell types. Previous findings show that the adult steady-state hematopoiesis relies on MLL1 activity, allowing for the maintenance of the hematopoietic stem cell (HSC) pool and promoting the proliferation of progenitors through expression of homeodomain and other transcription factors[30,41]. In the absence of Mll1, HSCs exit from quiescence to precociously enter cell cycle without an increase in their number, suggesting that HSC division is accompanied by differentiation and not self-renewal[30,41]. Additionally, MLL1 is critical for neural development and for post-natal differentiation of neurogenic stem cells[42,43]. However, in contrast to HSCs and muscle stem cells, Mll1-deficient neural stem cells are able to proliferate, but fail to activate Dlx2 to initiate the neuronal program[28].

Together, our data adds to and reinforces the understanding that MLL1 performs essential functions to regulate stem cell biology. Our work adds myogenesis to the list of stem cell dependent processes which require MLL1. Furthermore, we deciphered a new mechanism of satellite cell regulation that acts upstream of Pax7, and perturbation of this regulatory pathway leads to a profound deficit in satellite cell function. This work adds to our understanding of PAX7 transcription and the regulation of myogenesis upstream of PAX7 and may provide insight for future therapies targeting myogenic deficiencies or malignancies.

## Methods

**Mice and animal care**. *Mll1*[fl/fl] mice[30] and *Mll2*[fl/fl] mice[44] were crossed with *Rosa*[CE/+] mice[45] to obtain the *Rosa*[CE/+]:*Mll1*[fl/fl], *Rosa*[CE/+]:*Mll2*[fl/fl], and *Rosa*[CE/+]:*Mll1*[fl/fl]:*Mll2*[fl/fl] mice used for the in vitro experiments. Tamoxifen-inducible *Pax7*[CE/+]:*Mll1*[fl/fl]:*Rosa*[YFP] mice were generated by crossing *Pax7*[CE/+] mice[46] with *Gt(ROSA)26Sor*[tm3(CAG-EYFP)] (referred as *Rosa*[YFP] in the text) mice (Jackson Laboratory) and *Mll1*[fl/fl] mice. Mice were injected intraperitoneally with 100 μL of a 20 mg.mL[−1] tamoxifen solution dissolved in corn oil (Tmx, Sigma T5648) for 4 consecutive days, and maintained on tamoxifen diet (500 mg Tmx per kg diet, Teklad, Envigo). All the mice used in this study were males and females, from 6- to 16-week old, with mixed genetic background (129SV and C57BL/6). Housing, husbandry and all experimental protocols for mice used in this study were performed in accordance with the guidelines established by the University of Ottawa Animal Care Committee, which is based on the guidelines of the Canadian Council on Animal Care (CCAC). Protocols were approved by Animal Research Ethics Board (AREB) at the University of Ottawa.

**Muscle regeneration and histology**. Muscle injury was induced by injection of 50 μl of 10 μM cardiotoxin (CTX) from *Naja mossambica mossambica* (Latoxan, L8102) into the *tibialis anterior* (TA) muscle of isofluorane-anesthetized animals. For the double injury protocol, CTX was reinjected 21 days after the first injury. Seven, 21, and 42 days after CTX injections, mice were euthanized and TA muscles were harvested, weighed and embedded in OCT cooled with liquid nitrogen. For GFP visualization, mice were euthanized with a sodium pentobarbital solution (Euthanyl) and perfused with PBS before fixation with 4% paraformaldehyde (PFA) solution in PBS. Muscles were excised, further fixed into 0.1% PFA solution in PBS for 16 h, then transferred into ascending sucrose gradients (15%, 30% in PBS), and finally embedded and frozen in OCT. Embedded muscles were cryosectioned with 10 μm thickness.

For fiber counting and minimal Feret's diameter analysis, cryosections were washed with PBS, blocked in 5% goat serum, 2% BSA in PBS, stained with Wheat Germ Agglutinin Alexa647 conjugate for 1 h at room temperature and nuclei were counterstained with Hoechst (1:1000 in PBS). Images were taken on a Zeiss Axio

Observer.D1 inverted microscope equipped with an EC Plan-Neofluar 10 × /0.3 Ph1 M27 objective, and stitched together using Fiji software (http://fiji.sc/Fiji)[47]. Stitched pictures were analyzed using the SMASH software in MATLAB 2015a[48]. Total number of myofibers in each tissue was verified manually and more than 95% of myofibers were quantified across each muscle section.

**Fluorescence-activated cell sorting**. Satellite cells were prospectively isolated by flow cytometry using Alexa647-conjugated anti-α7-Integrin, biotin-conjugated anti-CD34 (streptavidin PE-Cy7 conjugate), and PE-conjugated anti-CD31, anti-CD45, anti-SCA1, and anti-CD11b antibodies[49]. Quiescent satellite cells were obtained from uninjured hind limb muscles while activated satellite cells were obtained from CTX-injured *tibialis anterior* and *gastrocnemius* muscles 3 days after the induced injury. The list of antibodies is provided in Supplementary Table 3. In brief, dissected muscles were minced in collagenase/dispase solution followed by dissociation using the gentleMACS Octo Dissociator with Heaters (Miltenyi Biotec). Cells were then filtered through a 70 μm nylon mesh and collected by centrifugation. Cell pellets were resuspended in FACS buffer (10% FBS, 5 mM EDTA in PBS) and incubated with the antibodies listed above. Unbound antibodies were washed with FACS buffer and cells were pelleted prior to resuspension in FACS buffer and a final filtration through a 30 μm filter. Satellite cells were sorted based on forward scatter and side scatter profiles, followed by negative lineage selection in PE (CD11b−, SCA1−, CD45−, CD31−) and positive selection in Alexa647 (α7-Integrin+) and PE-Cy7 (CD34+). Flow cytometry was performed using a MoFlo XDP (Beckman Coulter). The sequential gating strategy is described in Supplementary Fig. 9.

**Primary myoblast isolation and culture**. Primary myoblasts were derived from hind limb muscles of 5–6-week-old mice by magnetic cell separation (MACS)[50]. In brief, muscle dissociation and cell filtration was performed as described above for FACS. Cell pellets were resuspended in FACS buffer and incubated with biotin-conjugated lineage antibodies (CD11b−, SCA1−, CD45−, CD31−), followed by incubation with streptavidin microbeads for negative lineage selection. Cell suspensions were loaded onto a LD column (Miltenyi Biotec) in a magnetic field of a VarioMACS separator (Miltenyi Biotec) and rinsed with FACS buffer. Column flow-through was collected and then stained with biotin-conjugated anti-α7-Integrin antibody, followed by incubation with streptavidin microbeads, to purify the satellite cell-derived myoblasts. Cells were loaded onto a MS column (Miltenyi Biotec) and washed three times with FACS buffer. Flow-through containing unlabeled cells was discarded, and the column was flushed to collect positively labeled satellite cell-derived myoblasts. Myoblasts were cultured on collagen-coated dishes in Ham's F10 medium (Wisent, 318-051-CL) supplemented with 20% FBS, 1% penicillin/streptomycin, and 5 ng.mL[−1] of basic FGF (Millipore, GF003AFMG).

For in vitro knockout experiments, myoblasts were treated with 500 nM (Z)−4-Hydroxytamoxifen (4OHT, resuspended in ethanol (Sigma, H7904-5MG)), for 7 days followed by 2 days of culture in the absence of 4OHT prior to collection of cells for analysis. Differentiation was triggered in Ham's F10:DMEM 1:1 (Wisent, 319-016-CL) supplemented with 5% horse serum, and 1% penicillin/streptomycin.

**Primary myoblast transfection**. To obtain the *Pax7*-overexpressing primary myoblasts, *Pax7* cDNA was cloned into the pHAN backbone (with puromycin resistance driven from a distinct SV40 promoter)[13]. Control virus expressed puromycin resistance alone. Retrovirus was produced by transfection of Phoenix-ECO cells (ATCC CRL-3214) with Lipofectamine 2000 (ThermoFisher 11668019) according to manufacturer's instructions. After 48 h, 1 ml of viral supernatant was used to infect primary myoblasts. After 48 h, infected myoblasts were selected with 3 μg.mL[−1] of puromycin for 5 days and maintained on 0.5 μg.mL[−1] of puromycin.

For transient knockdown experiments, primary myoblasts were transfected twice with a 24-h interval, with 5 nM of a scrambled universal negative control siRNA or with 5 nM of siRNA targeting *Mll1*, *Mll2*, *Set1a*, or *Set1b* (TriFECTa™ Dicer-Substrate RNAi kit, IDT), using Lipofectamine RNAiMax (Invitrogen). Transfected myoblasts were collected 24 h after the second transfection for RNA extraction.

**RT-qPCR and gene expression analysis**. For primary myoblasts, RNA was extracted using the Nucleospin RNA II kit (Macherey-Nagel), according to the manufacturer's instructions. For satellite cells, RNA was isolated using Arcturus Picopure RNA extraction kit (Thermo Fisher Scientific). cDNA was synthesized using SuperScript III First-Strand Synthesis System (Invitrogen) by following the manufacturer's protocol. Real-time quantitative PCR reactions were performed using iQ SYBR Green Supermix (Bio-Rad) and the CFX96 Real-Time PCR Detection System (Bio-Rad). A list of the primers used can be found in Supplementary Table 1.

For microarray analysis, RNA was purified as described above. Samples were hybridized to Affymetrix MoGene 1.0 ST chipsets (Affymetrix). Raw data were RMA normalized using the Bioconductor R package. The RMA normalized data was log2 transformed to determine fold change in gene expression. The following criteria were used to derive Table 1: for downregulated genes, we chose genes exhibiting a log-fold change of greater than one (that is, twofold cut-off) and

Affymetrix signal >7 for the higher signal; for upregulated genes, 10 most highly upregulated genes with Affymetrix signal >8 for the higher signal were selected; for myogenic genes, we selected genes with known roles in myogenesis with emphasis on homeodomain containing transcription factors, and not listed in other sections of table; for MLL1 target genes, *Hox* genes and few genes previously purported to be regulated by MLL1 were selected with Affymetrix signal >7. All Affymetrix probe set values are provided in Supplementary Data file 1. Microarray data are available from the Gene Expression Omnibus, National Center for Biotechnology Information (https://www.ncbi.nlm.nih.gov/geo/) under series GSE108339. Gene ontology analysis were performed using g:profiler (https://biit.cs.ut.ee/gprofiler/gost), and FIMO was employed to scan the Jaspar database for transcription factor binding sites within the *Pax7* locus.

**Protein extraction and western blotting**. Primary myoblasts were lysed into RIPA buffer in the presence of EDTA-free Protease Inhibitor Cocktail (Roche). Histones were extracted from $2 \times 10^6$ primary myoblasts using the Histone Extraction Kit (ab113476, Abcam), according to the manufacturer's instructions. Equal amounts of proteins or histone extracts were resolved on SDS-PAGE and transferred onto PVDF membranes. Membranes were blocked using 5% non-fat dry milk in TBS-Tween 0.1% for 1 h at room temperature followed by incubation with primary antibodies overnight at 4 °C. After four washes in TBS-Tween 0.1%, membranes were incubated 1 h with HRP-conjugated secondary antibodies. After four more washes, immunoblots were developed by enhanced chemiluminescence. A list of primary antibodies is available in Supplementary Table 3. Full scans of the immunoblots are available in Supplementary Fig. 8.

**Immunofluorescence**. Primary myoblasts and muscle cross-sections were fixed 5 min in 4% PFA solution in PBS, permeabilized in 0.1 M Glycine, 0.1% Triton X-100 in PBS for 10 min and blocked in either 5% horse serum, 2% BSA in PBS, or in 5% goat serum, 2% BSA, MOM blocking reagent 1:40, in PBS for 1 h at room temperature. Primary antibodies were incubated overnight at 4 °C. After three washes in PBS, secondary antibodies were incubated in blocking buffer for 1 h at room temperature. Finally, muscle cross-sections were washed three times in PBS, nuclei were counterstained with DAPI (1:1000 in PBS), and slides were mounted with Permafluor. A list of primary and secondary antibodies is available in Supplementary Table 3. Images of immunostaining were taken on a Zeiss Axio Observer.D1 inverted microscope equipped with either a Plan-Apochromat 20×/0.8 M27 objective or a Plan-Apochromat 63×/1.4.Oil DIC M27 objective. Images were processed and analyzed with Zen and FIJI software.

**Chromatin immunoprecipitation**. Primary myoblasts were cross-linked with 1% formaldehyde for 10 min at room temperature. Glycine was added to the fixing solution and incubated for 5 min. Cells were lysed in lysis buffer (50 mM Tris-HCl pH 8.0, 10 mM EDTA, 0.5% SDS) and chromatin was sheared using a Covaris sonicator. The sheared DNA was centrifuged at $13,000 \times g$ for 5 min, and the remaining chromatin was diluted with an equal volume of dilution buffer (20 mM Tris-HCl pH 8.0, 1.2 mM EDTA, 1.1% Triton X-100, 200 mM NaCl). Chromatin was quantified using Bradford, and 500 μg of chromatin was used per chromatin immunoprecipitation (ChIP). Chromatin was pre-cleared for 30 min using protein A/G magnetic beads (30 μL per ChIP; Invitrogen). Antibodies (available in Supplementary Table 3) were added to the chromatin and incubated overnight with rotation at 4 °C. For transcription factor ChIP, 2 μg of anti-MLL1 N-terminal antibody or normal mouse IgG were used, and 2 μg of rabbit anti-mouse antibody was added to the reaction to amplify the signal. For histone modification ChIP, 5 μg of anti-Histone H3, anti-Histone H3K4me3 antibodies or normal rabbit IgG were used. The following day, protein A/G magnetic beads (30 μL per ChIP) were added to the lysate and incubated for 3 h with rotation at 4 °C. Beads were then washed three times with wash buffer (20 mM Tris-HCl pH 8.0, 150 mM NaCl, 2 mM EDTA, 0.1% SDS, 1% Triton X-100) for 5 min and two times with TE buffer for 5 min. DNA was eluted twice with freshly made elution buffer (0.1 M NaHCO₃, 1% SDS) for 30 min. Eluates were pooled, and proteinase K was added to the chromatin and incubated overnight at 65 °C. DNA was purified by phenol/chloroform extraction.

For ChIP in satellite cells, prospectively isolated satellite cells (CD31−, Cd11b−, SCA1−, CD45−, α7-Integrin+) were cross-linked and sonicated as described above. In total, $10^5$ cells were used per ChIP, and 5% input was kept. Primary antibodies (2 μg of anti-H3K4me3 or rabbit IgG per ChIP) were pre-incubated with magnetic A/G beads (15 μL per ChIP) for 3 h with rotation at 4 °C. The antibody/bead complexes were incubated with chromatin for 2 h with rotation at 4 °C. Beads were washed as described above. Elution, reverse cross-link and proteinase K digestion were performed in a single step for 2 h at 65 °C, using Micro-ChIP Elution buffer (20 mM Tris-HCl pH7.5, 50 mM NaCl, 5 mM EDTA, 1% SDS, 50 μg.mL⁻¹ proteinase K). DNA was purified using the Nucleospin Gel and PCR Cleanup with NTB buffer (Macherey-Nagel).

DNA enrichment was quantified by qPCR using the primers in Supplementary Table 2.

**Statistical analysis**. Statistical evaluation was performed using the Student's *t*-test tests to calculate differences between two groups and either one-way or two-way

ANOVA with post hoc test for multiple comparisons (Graphpad Prism®). Data are presented as mean ± S.E.M. and p-value < 0.05 was considered as statistically significant. Throughout the paper, level of significance is indicated as follows: *$p \leq 0.05$, **$p \leq 0.01$, ***$p \leq 0.001$.

**Reporting summary**. Further information on research design is available in the Nature Research Reporting Summary linked to this article.

## Data availability
Microarray data have been deposited in the Gene Expression Omnibus under the accession code GSE108339. The raw images for the immunoblots are provided in Supplementary Fig. 8. The source data underlying Figs. 1a-c, e, 2c, 3b, d-g, 4b-f, 5d-f, 6b-c, e, and Supplementary Figs. 2a, c, 3c, 4b, e-f, 5b-e, g, 7c, are provided in the Source Data file. All other data supporting the findings of this study are available from the corresponding author on reasonable request. A reporting summary for this Article is available as a Supplementary Information file.

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

## Acknowledgements

The authors thank Drs. Alex Lin and Jeffrey Dilworth for critical reading of the paper. We thank Jennifer Ritchie for animal husbandry, Fernando Ortiz for FACS, StemCore laboratories for microarray experiments, and Natasha Mercier and Hong Ming for technical assistance. G.C.A. was supported by master and doctoral fellowships from the Canadian Institutes of Health Research. C.E.B. is supported by postdoctoral fellowships from the Ontario Institute for Regenerative Medicine and the French Muscular Dystrophy Association (AFM)-Téléthon. J.S. is supported by a doctoral fellowship from the Canadian Institutes of Health Research. M.A.R. holds the Canada Research Chair in Molecular Genetics. These studies were carried out with support of grants to M.A.R. from the US National Institutes for Health [R01AR044031], the Canadian Institutes of Health Research [FDN-148387], the Muscular Dystrophy Association (USA), and the Stem Cell Network.

## Author contributions

G.C.A., C.E.B. and M.A.R. designed the experiments. G.C.A., C.E.B., M.-C.S., J.S. and C.J.P. conducted the experiments. G.C.A., C.E.B., M.-C.S., J.S., C.J.P. and M.A.R. analyzed the results. P.E. and A.F.S. provided *Mll1* and *Mll2* mutant mice. G.C.A., C.E.B., M.-C.S. and M.A.R. wrote the original draft. C.E.B. and M.A.R. wrote, reviewed, and edited the final paper. M.A.R. provided financial support.

## Additional information

**Competing interests:** The authors declare no competing interests.

