## [Peer Review File · Nature Communications]

Reviewers' comments:

Reviewer #1 (Remarks to the Author):

Work stemming from the same group had previously shown that PAX7 recruits MLL1 and MLL2 to epigenetically activate target genes. In this report, the authors used genetically ablated Mll1 and Mll2 flox alleles, and demonstrated that MLL1 is required for Pax7 transcription in activated but not quiescent satellite cells. Mll1 cKO myoblasts show impaired proliferation capacity, yet they retain their differentiation potential. In vivo, Mll1 cKO satellite cells seem to have no phenotype in a steady state. Upon injury-induced activation, however, the mutant cells show decreased proliferation and self-renewal. Using a combination of experiments the authors show that Mll2 is dispensable whereas Mll1 is required for Pax7 transcription in cultured cells. However, the transcription of Myf5 -which is sharply down in Mll1 KO cells- is regulated via PAX7 and not MLL1, as the early experiments suggested.

Overall the experiments are thoroughly performed and analyzed, although some of the conclusions would require further experimental support (see comment below). This study does not provide any major conceptual or technical advance, nor does it uncover any unexpected function as it analyses previously identified molecules in the muscle system. Nevertheless, it clarifies the functional relationship between these proteins and -for at least a part of the study- it uses conditional knock-out models to address the function of Mll1 in vivo.

Comments:

1/ A methyltransferase activity of MLL1 is the trimethylation of H3K4. By ChIP-qPCR on Mll1 KO cultured primary myoblasts, this study shows decreased H3K4me3 occupancy in the Pax7 and Myf5 loci. However, since Myf5/Pax7 are downregulated in the mutant cells, decreased H3K4me3 is somehow expected. So, although this experiment is well performed, it does not demonstrate that the H3K4me3 driven by MLL1. Therefore, the conclusion on page 7 that "Thus, our results indicate that MLL1 regulates both Pax7 and Myf5 expression through deposition of H3K4me3 at their regulatory elements in primary myoblasts" is not entirely justified.

Moreover, overexpression of Pax7 in Mll1 cKO myoblasts restored H3K4me3 enrichment at the Myf5 promoter in the absence of MLL1 (Fig 4c). Doesn't this experiment demonstrate that MLL1 is not required for H3K4me3 requirement? In addition, O/E of Pax7 did not increase H3K4me3 at the Myf5 locus, making less probable the idea that MLL1 3-methylates the Myf5 locus (Fig 2).

*Also, linked to the sentence cited above, according to the results of this study, MLL1 regulates Pax7 but not Myf5 expression. Please, rephrase

2/ A large number of data comes from cultured primary myoblasts. For a more complete picture of the role of MLL1 in vivo a transcriptomic analysis of quiescent and activated mutant cells should be performed.

3/ page 11: Since Mll1 deletion results in a strong downregulation of PAX7, this marker cannot be used to count satellite cells in the cKO muscle. Please. Use another satellite cell marker like M-cadherin or Syndecan-4.

4/ The sentence "Since loss of the Pax7 transcription factor results in a complete cell cycle arrest" is an over-statement. Even Pax7 null animals, albeit smaller, have some muscle and even some satellite cells at birth. Please, rephrase.

5/ Please, show the levels of the Mll1 and Mll2 transcripts in the microarrays.

Note for the authors: The Pax7-CT2 knock-in, knock-out mice used (Fan lab) seem to have a phenotype on their own. In addition, anecdotally, this phenotype is aggravated on the presence of Tamoxifen. So, the appropriate control for the in vivo studies would have been Tamoxifen-treated Pax7^{CE/+}; Mll1^{+/+}; RosaYFP.

Reviewer #2 (Remarks to the Author):

In this manuscript entitled, "MLL1 is Required for Pax7 Transcription and Satellite Cell Self-Renewal", Addicks, et al., analyze the role of MLL1 in regulating Pax7 expression in primary myoblasts. The authors first show by microarray and RT-qPCR analyses a significant reduction in the expression of the well-known Pax7 target gene, Myf5 in Mll1 cKO but not Mll2 cKO primary myoblasts. They further demonstrate that MLL1 is bound at the Myf5 and Pax7 promoters and that the loss of Mll1 binding in Mll1 cKO myoblasts results in a significant and moderate reduction in H3K4me3 levels at the Myf5 and Pax7 promoters, respectively. Using Mll1 cKO myoblasts, which reveal a significant reduction in Pax7 protein levels, the authors suggest that Pax7 is required for myoblast proliferation but not differentiation potential. They also found that overexpression of Pax7 in committed MLL1

cKO myoblasts restored H3K4me3 enrichment at the Myf5 promoter and subsequently the activation of this gene. Finally, Pax7 and Myf5 expression levels, cell proliferation, and self-renewal were examined following Mll1 deletion in quiescent and activated satellite cells.

Several analyses in the current manuscript remain preliminary and fail to convincingly support their main conclusions. Although the authors present an interesting finding that relates to the upstream regulation of Myf5 and myoblast proliferation that is mediated by Mll1 regulation of Pax7, the current study falls short in providing sufficient insight into the mechanism by which MLL1 is regulating Pax7 expression levels. There is also insufficient data to directly connect MLL1 regulation of Pax7 to the causal effects of Mll1 depletion in the current manuscript that include the significant decrease of Myf5 gene activation, myoblast proliferation, and satellite cell proliferation. Moreover, since the authors previously identified that Pax7 recruits MLL2 to activate target genes including Myf5 in quiescent satellite cells and primary myoblasts (McKinnell et al., 2007), we find that the current manuscript does not significantly advance their previous findings and thereby lacks novelty.

Major comments:

1. In relation to the investigation of the mechanism by which MLL1 regulates Pax7 expression, the authors should investigate the response elements found overlapping with the MLL1 binding sites within the Pax7 promoter to deduce the potential TFs that are recruiting MLL1 to regulate Pax7 expression levels. Also, as mentioned by the authors on page 8, the immunostaining analyses and Western blot analyses of Pax7 in the Mll1 cKO that are included in Fig. 3 reveal a significant decrease in Pax 7 protein levels, which appears to be significantly more pronounced than would be expected based on their RT-qPCR analyses of Pax7 expression levels and their ChIP findings showing only a moderate reduction in the levels of H3K4me3 at the Pax7 promoter in the Mll1 cKO conditions. These findings suggest that Mll1 regulation of Pax7 is independent of the HMT activity of Mll1, a possibility that is mentioned in the Discussion section. This point should be addressed with a further assessment of the mechanism by which Mll1 is regulating Pax7 protein levels since this is the primary finding of the current manuscript. For example, other cofactor functions of the MLL1 complex (other than its HMT activity), should be assessed by targeting the catalytic domain of MLL1 followed by a subsequent assessment of Pax7 regulation.
2. The immunostaining results examining the efficiency of Mll1 and Mll2 deletion were not included in Fig. 1a,b as reported in the text on pg. 5.
3. In Figure 1, the authors should confirm the single and double Mll1 and Mll2 knockdown conditions by including the described immunostaining analysis that we found to be missing in Fig. 1 or immunoblot analyses for MLL1 and MLL2. The authors claim that culture myoblasts are committed and exhibit limited plasticity to explain why the microarray data shows little change in the expression of Mll1 target genes. Nonetheless, it should be ruled out that the expression levels of the analyzed Mll1 target genes are not affected because of negligible decreases in Mll1 and Mll2 protein levels.

4. In Supplementary Fig. 2a, the authors observe a significant reduction in Mll1 binding at the Pax7 promoter region in the Mll1 cKO myoblasts. However, in Figure 2, they show only a moderate loss of H3K4me3 at the Pax7 promoter, which corresponds with their data in Fig. 1a showing that Pax7 expression levels are significantly, but moderately affected in the Mll1 cKO primary myoblasts. The authors should normalize their H3K4me3 ChIP results with the amount of histone H3 immunoprecipitated, which could help in addressing the above point. The authors should also test whether another HMT is functioning to maintain the H3K4me3 levels at the Pax7 promoter following the loss of Mll1.
5. In Fig. 2, the authors should examine whether the global levels of H3K4me3 are changing by analyzing histone extracts in immunoblot analyses, to determine whether the loss in H3K4me3 levels in the Mll1 cKO conditions are gene-specific or global. Should this data suggest that the effects on H3K4me3 levels is gene-selective than this data could provide further support for their microarray data that showed little change in the total number of genes whose expression was affected by Mll1 depletion.
6. In Fig. 2, the authors should include H3K4me3/H3 ChIP data for the +1.1kb region of Pax7, particularly since this region was found to show a significant decrease in MLL1 binding in the Mll1 cKO analyses in Supplementary Fig. 2. It is also unclear why the -27 kb and -3.5 kb regions of Pax7 were left out in Fig. 2 but included in Supplementary figure 2. These figures should match-up or else clarification should be provided as to why these genomic regions were left out of the analyses in Fig. 2.
7. In Figure 2, the authors should mention why they are identifying a significant loss of MLL1 binding that corresponds with significant decreases in H3K4me3 levels at the genomic regions +11.5 and +11.7 kb that are significantly downstream of the TSS/promoter region of Pax7.
8. In Fig. S2b, the authors conclude that MLL1 KO does not result in accumulation of H3K27me3 and subsequent downregulation through the deposition of this repressive mark. The authors should demonstrate the success of their H3K27me3 ChIP assays by including a positive control gene that would be expected to reveal a change in H3K27me3 levels.
9. In Fig. 3, despite the significant decrease in Pax7 protein levels in the Mll1 cKO conditions, additional analyses including Pax7 depletion through independent means (ie. shRNA knockdown) should be performed to further support the claim that the loss of Pax7 affects proliferation since MLL1 depletion could be affecting proliferation through a mechanism that is independent of Pax7.
10. In the description of Fig. 3e, the authors claim on page 7 that Mll1 deletion results in a 3.5-fold increase (rather than decrease) in the number of Mll1 cKO myoblasts. Yet, the data in Fig.3e, f demonstrates that Mll1 deletion results in a significant decrease in the number of myoblasts.
11. For Fig. 3h, the corresponding figure legend should include how many days were allowed for differentiation prior to the immunostaining for MyHC.
12. In Fig. 4a, is the fold reduction in Pax7 expression comparable to what was observed in Fig. 1a in the control versus cKO cells? This information should be included in the text.

13. Western blot analyses are needed to examine the protein levels of Pax7 to accompany the RT-qPCR analyses in Fig. 4a to assess comparable levels in the Ctr versus cKO conditions.
14. In Fig. 4 of the current manuscript, Pax7 overexpression is found to be sufficient to restore Myf5 activation which is accompanied by Pax7-dependent increases in the H3K4me3 levels at the Myf5 promoter. However, while there is a significant increase in the number of myoblasts that proliferate following Pax7 overexpression, the total number of proliferating myoblasts is significantly lower as compared to the myoblasts that express Mll1. As the authors showed in their previous study (McKinnell et al., 2007), Pax7 through MLL2 interactions regulate Myf5 activation and thereby could explain the Pax7 rescue of H3K4me3 levels at the Myf5 promoter and Myf5 expression levels in the absence of Mll1. This possibility should be examined to extend the mechanistic insights of their current study by more specifically assessing MLL1/MLL2 redundancy.
15. In relation to Fig. 4d and based on the microarray analysis, how many MLL1-dependent genes are also direct Pax7 targets (similar to Fgfr4)? It would be helpful to provide this information when discussing this data and as basis for the claim that “loss of MLL1 in myoblasts results in changes to gene expression that are both dependent and independent to changes in Pax7 expression”.
16. In supplementary figure 5, the authors should confirm the single Mll1 knockdown conditions by including immunostaining or immunoblot analyses for Mll1.

Minor Comments:

1. In Fig. 1 or Fig. 3, the Myf5 protein levels should be examined by western blot analyses in the Mll1 cKO conditions.
2. There are multiple corrections in the text that are needed and have been outlined. This manuscript needs to be carefully reviewed and rewritten.
3. To better reflect the differences in decreased H3K4me3 deposition following MLL1 KO at the two different loci (Myf5 and Pax7), % change values should be included.
4. Statistical analysis is needed for the microarray data.
5. Fig. S3 is missing the graph legend (for the colored bar graphs).
6. The figure subpanel labeling (A, B, etc.) should be consistent in all the figures and with references in the text (upper vs. lower case).
7. Some of the methods subsections (ChIP as an example) need additional detail.

Reviewer #3 (Remarks to the Author):

Addicks et al. report that deletion of MLL1, a H3K4 methyltransferase, in satellite cells resulted in reduced H3K4me3 at both Pax7 and Myf5 promoters and impaired their expression. Mll1 cKO, but not Mll2 cKO, satellite cells displayed proliferation defects and MLL1^{-/-} mice had reduced satellite cell proliferation and self-renewal and impaired skeletal muscle regeneration after injury. Pax7 overexpression partially corrected the proliferation defect and fully restored Pax7 and Myf5 expression.

Overall, this is a well conducted and interesting study.

1. Table 1. In Mll1 cKO myoblasts, expression of 929 genes was increased and of 78 genes (15 coding mRNAs) decreased. Given the positive role exerted by H3K4me3 on transcription, it is likely that downregulation is an indirect effect of MLL1 deletion. CHIP-seq for MLL1 would identify direct versus indirect targets.

2. A GO analysis and discussion of upregulated genes in Mll1cKO myoblasts would be helpful.

3. Myogenin is not expressed in myoblasts and thus its expression is not expected to be affected by MLL1.

4. Figure 3. Comparable number of myoblasts are shown for control and Mll1cKO in Figure 3a. However, less than 20% of Mll1 cKO myoblasts are Pax7+. What is the identity of the remaining 80%? Do they retain other myoblasts-specific markers?

5. Figure 4c. As noted in the Discussion, it is surprising that CMV-Pax7 expression restored H3K4me3 at Myf5 in Mll1 cKO myoblasts. A CHIP-qPCR for Set1A/B in Mll1 cKO myoblasts would directly evaluate the authors' suggestion that this methyltransferase can substitute for Mll1.

6. Supplemental Figure 5. CHIP-qPCR for MLL1 at the Pax7 locus in quiescent and activated satellite cells would help in clarifying the seemingly distinct role of MLL1 in the two cell states.

7. Microarray datasets of Mll1 cKO and Pax7 cKO myoblasts should be compared to uncover differences and similarities.

Response to reviewers' comments

We are grateful to the reviewers for appraising our work and for providing detailed feedback and helpful comments for improvement. All of the points have been addressed. We hope sincerely that these modifications strengthen our manuscript and our conclusion that MLL1 is required for Pax7 expression and satellite cell function during skeletal muscle regeneration.

Reviewer #1 (Remarks to the Author)

Work stemming from the same group had previously shown that PAX7 recruits MLL1 and MLL2 to epigenetically activate target genes. In this report, the authors used genetically ablated Mll1 and Mll2 flox alleles and demonstrated that MLL1 is required for Pax7 transcription in activated but not quiescent satellite cells. Mll1 cKO myoblasts show impaired proliferation capacity, yet they retain their differentiation potential. In vivo, Mll1 cKO satellite cells seem to have no phenotype in a steady state. Upon injury-induced activation, however, the mutant cells show decreased proliferation and self-renewal. Using a combination of experiments the authors show that Mll2 is dispensable whereas Mll1 is required for Pax7 transcription in cultured cells. However, the transcription of Myf5 -which is sharply down in Mll1 KO cells- is regulated via PAX7 and not MLL1, as the early experiments suggested.

Overall the experiments are thoroughly performed and analyzed, although some of the conclusions would require further experimental support (see comment below). This study does not provide any major conceptual or technical advance, nor does it uncover any unexpected function as it analyses previously identified molecules in the muscle system. Nevertheless, it clarifies the functional relationship between these proteins and -for at least a part of the study- it uses conditional knock-out models to address the function of Mll1 in vivo.

We thank the reviewer for his positive comments. However, we would like to point out that our study provides the first example of a transcriptional regulator that is absolutely required for Pax7 expression and that phenocopies the *Pax7*^{-/-} mouse model. Loss of MLL1 abolishes the expression of Pax7, leading to a dramatic impairment of satellite stem cell proliferation and self-renewal during muscle regeneration. Therefore, we believe that this manuscript does provide significant new insight into understanding Pax7 regulation and the role played by MLL1.

Comments:

1/ A methyltransferase activity of MLL1 is the trimethylation of H3K4. By ChIP-qPCR on Mll1 KO cultured primary myoblasts, this study shows decreased H3K4me3 occupancy in the Pax7 and Myf5 loci. However, since Myf5/Pax7 are downregulated in the mutant cells, decreased H3K4me3 is somehow expected. So, although this experiment is well performed, it does not demonstrate that the H3K4me3 driven by MLL1. Therefore, the conclusion on page 7 that "Thus, our results indicate that MLL1 regulates both Pax7 and Myf5 expression through deposition of H3K4me3 at their regulatory elements in primary myoblasts" is not entirely justified. Moreover, overexpression of Pax7 in Mll1 cKO myoblasts restored H3K4me3 enrichment at the Myf5 promoter in the absence of MLL1 (Fig 4c). Doesn't this experiment demonstrate that MLL1 is not required for H3K4me3 requirement? In addition, O/E of Pax7 did not increase H3K4me3 at the Myf5 locus, making less probable the idea that MLL1 3-methylates the Myf5 locus (Fig 2). *Also, linked to the sentence cited above, according to the results of this study, MLL1 regulates Pax7 but not Myf5 expression. Please, rephrase.

Following the reviewer suggestions, the entire paragraph (entitled: MLL1 binds both Pax7 and Myf5 genomic loci) has been re-written, and this sentence is no longer in the updated version.

2/ A large number of data comes from cultured primary myoblasts. For a more complete picture of the role of MLL1 in vivo a transcriptomic analysis of quiescent and activated mutant cells should be performed.

Microarray on primary myoblasts revealed that only a very small number of genes varied following *Mll1* deletion (**Table 1, Fig. 1**). Among them, RT-qPCR confirmed that *Pax7*, *Myf5*, *Asb4*, *Six2*, *Fgfr4* and *Lbx1* were significantly decreased (**Fig. 1a, e**).

We have now tested the expression of all these genes in quiescent and activated satellite cells isolated from *Pax7^{CE/+}:Mll1^{fl/fl}:Rosa^{YFP}* mice treated or not with tamoxifen (**Supplementary Fig. 5b, d**). RT-qPCR showed that their pattern of expression was very similar between primary myoblasts and activated satellite cells (**Fig. 1e** and **Supplementary Fig. 5d**). In quiescent satellite cells, while some of the MLL1 target genes are downregulated, the expression of Pax7 and its target genes does not vary (**Supplementary Fig. 5b**).

Therefore, we addressed the question using RT-qPCR in quiescent and activated satellite cells, and we do not think that transcriptomic analysis would be more informative than the one performed on primary myoblasts.

3/ page 11: Since *Mll1* deletion results in a strong downregulation of PAX7, this marker cannot be used to count satellite cells in the cKO muscle. Please. Use another satellite cell marker like M-cadherin or Syndecan-4.

We agree with the reviewer that using Pax7 as a marker may not be a reliable way to quantify the number of satellite cells after *Mll1* deletion. Unfortunately, M-Cadherin and Syndecan-4 immunostaining on cryosections of regenerated muscle is challenging and our trials were unsuccessful. However, taking advantage of the RosaYFP reporter allele in our *Pax7^{CE/+}:Mll1^{fl/fl}:Rosa^{YFP}* mice, we were able to track the satellite cell population during regeneration (**Fig. 7**). As showed in **Fig. 7c** and specified in the manuscript, only very rare YFP+ satellite cells were observed. These cells were always found co-expressing Pax7 and their number corresponded to the quantification in **Fig. 6e**.

However, to fully address the reviewer's concerns regarding the loss of satellite cells in injured muscles of tamoxifen-treated *Pax7^{CE/+}:Mll1^{fl/fl}:Rosa^{YFP}* mice, the regenerated TA muscle was subjected to a second injury (**Supplementary Fig. 7**). While the control mice were able to properly regenerate, the tamoxifen-treated *Pax7^{CE/+}:Mll1^{fl/fl}:Rosa^{YFP}* mice completely failed to regenerate, exhibiting a phenotype very similar to the one observed after genetic ablation of Pax7 (**Supplementary Fig. 7e-g**) (Von Maltzahn *et al.*, 2013, PMID: 24065826). This experiment clearly confirms that *Mll1* deletion leads to the loss of satellite cells during muscle regeneration.

4/ The sentence "Since loss of the Pax7 transcription factor results in a complete cell cycle arrest" is an over-statement. Even Pax7 null animals, albeit smaller, have some muscle and even some satellite cells at birth. Please, rephrase.

We agree with the reviewer and we rephrased our sentence which now reads as "Since loss of the Pax7 transcription factor results in reduced proliferation [...]"

5/ Please, show the levels of the *Mll1* and *Mll2* transcripts in the microarrays.

As requested, we added the transcript levels of *Mll1* and *Mll2* in the **Table 1**. As expected, no significant differences were observed in *Mll1* and *Mll2* transcripts between control and *Mll1* cKO myoblasts in the microarray. However, the fold change for the “exon 3-exon 4” region of *Mll1* confirms the loss of the floxed region (**Supplementary Fig. 1d**).

Note for the authors: The Pax7-CT2 knock-in, knock-out mice used (Fan lab) seem to have a phenotype on their own. In addition, anecdotally, this phenotype is aggravated on the presence of Tamoxifen. So, the appropriate control for the *in vivo* studies would have been Tamoxifen-treated Pax7^{CE/+}; Mll1^{+/+}; Rosa^{YFP}.

We agree with the reviewer that the best control for the *in vivo* studies would have been the tamoxifen-treated Pax7^{CE/+}:Mll1^{+/+}:Rosa^{YFP} mice. When we started our breeding, we noticed that the Pax7CreERT2 (Fan lab, Pax7^{CE/+}) mice were smaller than the wild-type (Pax7^{+/+}) (see Figure A below), indeed suggesting that they may have a phenotype on their own. We also assessed the consequences of tamoxifen intraperitoneal injections (IPs) and diet on the body weight. Although the body weight of the Pax7^{+/+}:Mll1^{fl/fl} mice was significantly decreased after 3 weeks of tamoxifen diet, we did not observe significant effects on the body weight of Pax7^{CE/+}:Mll1^{fl/fl} mice (Figure A). Therefore, in order to simplify our breeding and minimise the number of animals used for the study following the principle of the 3Rs, all our experiments were performed on mice harboring both the Pax7CreERT2 (Pax7^{CE/+}) and the *Mll1*-floxed (Mll1^{fl/fl}) alleles. In addition, the phenotype of the Pax7^{CE/+}:Mll1^{fl/fl} is so prominent, it is clearly not due to the well-characterized Fan allele on its own.

Figure A: Body weight of 10 week old mice. At 6 week old, Pax7^{+/+} and Pax7^{CE/+} mice were injected intraperitoneally with tamoxifen solution for 4 consecutive days, and maintained on tamoxifen diet for another 3 weeks (Tmx (IP + diet)). At 10 weeks, their body weight was compared to the non-treated Pax7^{+/+} and Pax7^{CE/+} mice (NO Tmx) (n = 5 mice per genotype).

Reviewer #2 (Remarks to the Author)

In this manuscript entitled, “MLL1 is Required for Pax7 Transcription and Satellite Cell Self-Renewal”, Addicks, et al., analyze the role of MLL1 in regulating Pax7 expression in primary myoblasts. The authors first show by microarray and RT-qPCR analyses a significant reduction in the expression of the well-known Pax7 target gene, Myf5 in Mll1 cKO but not Mll2 cKO primary myoblasts. They further demonstrate that MLL1 is bound at the Myf5 and Pax7 promoters and that the loss of Mll1 binding in Mll1 cKO myoblasts results in a significant and moderate reduction in H3K4me3 levels at the Myf5 and Pax7 promoters,

respectively. Using Mll1 cKO myoblasts, which reveal a significant reduction in Pax7 protein levels, the authors suggest that Pax7 is required for myoblast proliferation but not differentiation potential. They also found that overexpression of Pax7 in committed MLL1 cKO myoblasts restored H3K4me3 enrichment at the Myf5 promoter and subsequently the activation of this gene. Finally, Pax7 and Myf5 expression levels, cell proliferation, and self-renewal were examined following Mll1 deletion in quiescent and activated satellite cells.

Several analyses in the current manuscript remain preliminary and fail to convincingly support their main conclusions. Although the authors present an interesting finding that relates to the upstream regulation of Myf5 and myoblast proliferation that is mediated by Mll1 regulation of Pax7, the current study falls short in providing sufficient insight into the mechanism by which MLL1 is regulating Pax7 expression levels. There is also insufficient data to directly connect MLL1 regulation of Pax7 to the causal effects of Mll1 depletion in the current manuscript that include the significant decrease of Myf5 gene activation, myoblast proliferation, and satellite cell proliferation. Moreover, since the authors previously identified that Pax7 recruits MLL2 to activate target genes including Myf5 in quiescent satellite cells and primary myoblasts (McKinnell et al., 2007), we find that the current manuscript does not significantly advance their previous findings and thereby lacks novelty.

We respectfully disagree with the conclusion of the reviewer. First, our study provides evidence that MLL1 is an upstream transcriptional regulator of Pax7. Moreover, our work demonstrates that MLL1 regulates Pax7 expression during satellite cell activation and therefore, it is required for satellite cell proliferation and self-renewal during skeletal muscle regeneration. We performed many additional experiments to reinforce our conclusions. Therefore, we believe that the findings are novel and important.

Major comments:

1. In relation to the investigation of the mechanism by which MLL1 regulates Pax7 expression, the authors should investigate the response elements found overlapping with the MLL1 binding sites within the Pax7 promoter to deduce the potential TFs that are recruiting MLL1 to regulate Pax7 expression levels.

We performed *in silico* analysis looking at the 5 Pax7 loci bound by MLL1 (Pax7 -0,2 kb; Pax7 +0,6 kb; Pax7 +1,1 kb; Pax7 +11,5 kb; Pax7 +11,7 kb). We used FIMO to scan the Jaspar database for transcription factor binding sites that are common between these loci and restricted our analyses to TF's that are expressed in satellite cells. We did not identify any transcription factor binding motif that is present in all 5 MLL1 binding sites. Our results revealed 5 TFs for which at least 3 binding sites were identified, located in 2 or more different loci:

Klf5 (Kruppel-like factor 5)	(5 binding motifs in 3 different loci)
Ewsr1 (Ewing sarcoma breakpoint region 1)	(3 binding motifs in 2 different loci)
Sp2	(4 binding motifs in 3 different loci)
Sp3	(3 binding motifs in 3 different loci)
Zfp263	(10 binding motifs in 3 different loci)

An interaction between Ewsr1 and Mll1 has been reported using BioID (Elzi DJ *et al.*, 2014, PMID: 24999758). We added these results in the discussion section.

Also, as mentioned by the authors on page 8, the immunostaining analyses and Western blot analyses of Pax7 in the Mll1 cKO that are included in Fig. 3 reveal a significant decrease in Pax 7 protein levels, which appears to be significantly more pronounced than would be expected based on their RT-qPCR analyses of Pax7 expression levels and their CHIP findings showing only a moderate reduction in the levels of H3K4me3 at the Pax7 promoter in the Mll1 cKO conditions. These findings suggest that Mll1 regulation of Pax7 is independent of the HMT activity of Mll1, a possibility that is mentioned in the Discussion section. This point should be addressed with a further assessment of the mechanism by which Mll1 is regulating Pax7 protein levels since this is the primary finding of the current manuscript. For example, other cofactor functions of the MLL1 complex (other than its HMT activity), should be assessed by targeting the catalytic domain of MLL1 followed by a subsequent assessment of Pax7 regulation.

We agree that our results suggest that MLL1 regulates Pax7 independently of its HMT activity. However, MLL1 also regulates Pax7 at the transcriptional level, as shown by RT-qPCR and by ChIP, and we chose to focus on this type of regulation in the present manuscript. We agree that the post-transcriptional regulation of Pax7 by Mll1 is an interesting topic, but we believe that it falls outside of the scope of our present study.

2. The immunostaining results examining the efficiency of Mll1 and Mll2 deletion were not included in Fig. 1a,b as reported in the text on pg. 5.

To address this oversight, Western blot and immunostaining were added in the **Supplementary Fig. 1b, c** for MLL1 and confirmed the loss of MLL1 protein upon 4-hydroxytamoxifen (4-OHT) treatment. Despite the use of two different antibodies for MLL2 (anti-Trx2 antibody, Bethyl, A300-113A; anti-MLL2 antibody, Abnova, PAB14066), Western blot and immunostaining did not work and the loss of MLL2 was only confirmed by RT-qPCR (**Fig. 1b**).

3. In Figure 1, the authors should confirm the single and double Mll1 and Mll2 knockdown conditions by including the described immunostaining analysis that we found to be missing in Fig. 1 or immunoblot analyses for MLL1 and MLL2. The authors claim that culture myoblasts are committed and exhibit limited plasticity to explain why the microarray data shows little change in the expression of Mll1 target genes. Nonetheless, it should be ruled out that the expression levels of the analyzed Mll1 target genes are not affected because of negligible decreases in Mll1 and Mll2 protein levels.

As requested, we performed Western blot and/or immunostaining of MLL1 to confirm its significant loss upon 4-OHT treatment (**Supplementary Fig. 1b, c**). Our results rule out the possibility that the little change observed in MLL1 target gene expression is due to a negligible decrease of MLL1 protein in primary myoblasts.

4. In Supplementary Fig. 2a, the authors observe a significant reduction in Mll1 binding at the Pax7 promoter region in the Mll1 cKO myoblasts. However, in Figure 2, they show only a moderate loss of H3K4me3 at the Pax7 promoter, which corresponds with their data in Fig. 1a showing that Pax7 expression levels are significantly, but moderately affected in the Mll1 cKO primary myoblasts. The authors should normalize their H3K4me3 ChIP results with the amount of histone H3 immunoprecipitated, which could help in addressing the above point.

As suggested by the reviewer, we have conducted new ChIP experiments in order to normalize the H3K4me3 enrichment to the total amount of histone H3 (**Fig. 2c** and **Fig. 4d**). These new data are

essentially identical with our previous results, showing that H3K4me3 enrichment is moderately decreased at the *Pax7* promoter after *Mll1* deletion (**Fig. 2c**).

The authors should also test whether another HMT is functioning to maintain the H3K4me3 levels at the *Pax7* promoter following the loss of *Mll1*.

See Reviewer #3 point 5 for the response where we address possible redundancy between different HMTs.

5. In Fig. 2, the authors should examine whether the global levels of H3K4me3 are changing by analyzing histone extracts in immunoblot analyses, to determine whether the loss in H3K4me3 levels in the *Mll1* cKO conditions are gene-specific or global. Should this data suggest that the effects on H3K4me3 levels is gene-selective than this data could provide further support for their microarray data that showed little change in the total number of genes whose expression was affected by *Mll1* depletion.

To address this issue, we have performed histone extraction in *Mll1* cKO and control myoblasts. We have analyzed the proportions of each histones following *Mll1* deletion (**Supplementary Fig. 2c, 2d**). No change was observed. We have also conducted Western blot analysis for H3K4me3 in each cell line, showing that *Mll1* deletion does not trigger changes in global H3K4me3 levels compared to the control (**Supplementary Fig. 2d**). Together, these new data confirm that the effects are rather gene-selective than global.

6. In Fig. 2, the authors should include H3K4me3/H3 ChIP data for the +1.1kb region of *Pax7*, particularly since this region was found to show a significant decrease in *MLL1* binding in the *Mll1* cKO analyses in Supplementary Fig. 2. It is also unclear why the -27 kb and -3.5 kb regions of *Pax7* were left out in Fig. 2 but included in Supplementary figure 2. These figures should match-up or else clarification should be provided as to why these genomic regions were left out of the analyses in Fig. 2.

As requested, we repeated all the H3K4me3/H3 ChIPs and included the +1,1 kb region in our new ChIP sets (**Fig. 2c**). The -27 and -3.5 kb regions were left out in the first submission because we did not observe any *Mll1* binding at these regions. In the new version, we kept the -27kb region. **Fig. 2c** and **Supplementary Fig. 2a** now match-up.

7. In Figure 2, the authors should mention why they are identifying a significant loss of *MLL1* binding that corresponds with significant decreases in H3K4me3 levels at the genomic regions +11.5 and +11.7 kb that are significantly downstream of the TSS/promoter region of *Pax7*.

We added a sentence to clarify the analysis of the genomic regions +11.5kb and +11.7kb in the manuscript. In mouse embryonic stem cells, Marson *et al.* (2008, PMID: 18692474) showed that *Pax7* locus possesses two peaks of H3K4me3 enrichment downstream from the TSS (see below, Figure B). Thus, we included the analysis of both regions in our study.

Figure B: H3K4me3 enrichment at the *Pax7* locus in mouse embryonic stem cells. In mouse ES cells, H3K4me3 is enriched near the *Pax7* transcription start site (TSS; -0.2, +0.6, +1.1) and downstream from the *Pax7* TSS (GEO accession GSE11724, Marson *et al.*, 2008, PMID: 18692474).

8. In Fig. S2b, the authors conclude that MLL1 KO does not result in accumulation of H3K27me3 and subsequent downregulation through the deposition of this repressive mark. The authors should demonstrate the success of their H3K27me3 ChIP assays by including a positive control gene that would be expected to reveal a change in H3K27me3 levels.

We agree with the reviewer that we should have included a better positive control for the H3K27me3 ChIP-qPCR. Figure C (below) shows that the overexpression of exogenous *Pax7* leads to increased H3K27me3 enrichment at the *Pax7* locus, correlated with a decreased expression of endogenous *Pax7* transcripts. This data confirms that the H3K27me3 ChIP assay works and it also demonstrates that the H3K27me3 mark is not changed in *Mll1* cKO myoblasts. However, for a matter of simplicity, we decided to remove the H3K27me3 data from the present manuscript, as it does not change the main conclusions of our research.

Figure C: H3K4me3 and H3K27me3 enrichment at the *Pax7* locus in control and *Mll1* cKO myoblasts overexpressing a CMV-empty plasmid or a CMV-Pax7 plasmid. (a) RT-qPCR showing that exogenous *Pax7* expression results in a downregulation of endogenous *Pax7* expression. (b) Exogenous *Pax7* overexpression leads to a decrease in H3K4me3 enrichment while (c) H3K27me3 is highly increased at the *Pax7* locus.

9. In Fig. 3, despite the significant decrease in *Pax7* protein levels in the *Mll1* cKO conditions, additional analyses including *Pax7* depletion through independent means (ie. shRNA knockdown) should be

performed to further support the claim that the loss of Pax7 affects proliferation since MLL1 depletion could be affecting proliferation through a mechanism that is independent of Pax7.

We agree with the reviewer that *Mll1* depletion can affect proliferation independently of Pax7. Indeed, Pax7-overexpressing *Mll1* cKO myoblasts still exhibit defects in proliferation compared to the Pax7-overexpressing control myoblasts (**Fig. 4f**). Therefore, it is most likely that the proliferation impairment observed in *Mll1* cKO myoblasts is due to the loss of MLL1 rather than the loss of Pax7. We clarified our description of the results and our conclusions in the manuscript.

10. In the description of Fig. 3e, the authors claim on page 7 that Mll1 deletion results in a 3.5-fold increase (rather than decrease) in the number of Mll1 cKO myoblasts. Yet, the data in Fig. 3e, f demonstrates that Mll1 deletion results in a significant decrease in the number of myoblasts.

After 6 days of culture, there is actually a 3.5-fold increase in the number of *Mll1* cKO myoblasts compared to a 40-fold increase in the number of control myoblasts, overall resulting in a significant decrease in proliferation in the *Mll1* cKO myoblasts. We have re-written the sentence to clarify our experimental conclusions as follows: “After 6 days of culture, *Mll1* cKO myoblasts underwent a modest 3.5-fold expansion while control myoblasts expanded by 40-fold”.

11. For Fig. 3h, the corresponding figure legend should include how many days were allowed for differentiation prior to the immunostaining for MyHC.

Immunostaining was performed after 6 days of differentiation. We have added the number of days in the figure legend.

12. In Fig. 4a, is the fold reduction in Pax7 expression comparable to what was observed in Fig. 1a in the control versus cKO cells? This information should be included in the text.

Indeed, Pax7 expression in the control versus *Mll1* cKO myoblasts is comparable in **Fig. 1a** and **Fig. 4b**. We have specified it in the text, and we have cut the y axis to make the figure clearer for the RT-qPCR results.

13. Western blot analyses are needed to examine the protein levels of Pax7 to accompany the RT-qPCR analyses in Fig. 4a to assess comparable levels in the Ctr versus cKO conditions.

We have conducted new Western blot and immunostaining to analyze MLL1 and Pax7 expression in control and *Mll1* cKO myoblasts transduced with the CMV-empty plasmid or the CMV-Pax7 plasmid (**Fig. 4a** and **Supplementary Fig. 4c, d**). Our results confirm that *Mll1* deletion leads to decreased Pax7 expression in the *Mll1* cKO myoblasts transduced with the empty plasmid compared to the control. We also show that Pax7 protein is present at similar levels in the two Pax7-overexpressing cells (control vs *Mll1* cKO myoblasts).

14. In Fig. 4 of the current manuscript, Pax7 overexpression is found to be sufficient to restore Myf5 activation which is accompanied by Pax7-dependent increases in the H3K4me3 levels at the Myf5 promoter. However, while there is a significant increase in the number of myoblasts that proliferate following Pax7 overexpression, the total number of proliferating myoblasts is significantly lower as compared to the myoblasts that express Mll1. As the authors showed in their previous study (McKinnell et al., 2007), Pax7 through MLL2 interactions regulate Myf5 activation and thereby could explain the Pax7 rescue of H3K4me3 levels at the Myf5 promoter and Myf5 expression levels in the absence of Mll1. This

possibility should be examined to extend the mechanistic insights of their current study by more specifically assessing MLL1/MLL2 redundancy.

See Reviewer #3 point 5 for the response where we address possible redundancy between different HMTs.

15. In relation to Fig. 4d and based on the microarray analysis, how many MLL1-dependent genes are also direct Pax7 targets (similar to *Fgfr4*)? It would be helpful to provide this information when discussing this data and as basis for the claim that “loss of MLL1 in myoblasts results in changes to gene expression that are both dependent and independent to changes in Pax7 expression”.

As requested, we performed bioinformatics analysis to identify Pax7 and/or MLL1 target genes. Pax7 target genes were identified by intersecting *Mll1* cKO microarray data with microarray data from Pax7-overexpressing cells (Soleimani *et al.*, 2012, PMID: 22609161). Out of the 21 mRNA-coding genes that are down-regulated following *Mll1* deletion, we identified 9 genes (43%) that are bound by MLL1 in myoblasts (Cheng *et al.*, 2014, PMID: 24656132), and 4 of these (44%) are also Pax7 targets. From the 281 genes that are up-regulated in *Mll1* cKO myoblasts, 54 are bound by MLL1, 31 are bound by Pax7, and 10 are bound by both. We fully discuss these results in the revised manuscript.

16. In supplementary figure 5, the authors should confirm the single Mll1 knockdown conditions by including immunostaining or immunoblot analyses for Mll1.

MLL1 protein expression could not be tested by immunostaining on cryosections as both anti-MLL1 and anti-Pax7 antibodies are mouse IgG1 isotype. Immunoblot was not an option as the deletion is specific to satellite cells, and satellite cells are present in insufficient numbers to perform protein expression analyses. However, we performed additional experiments demonstrating that MLL1 target genes were downregulated in *Mll1*-deficient satellite cells (**Supplementary Fig. 5b, d**) as observed in *Mll1*-deficient primary myoblasts. Finally, the muscle phenotype is consistent with our observations in primary myoblasts (**Fig. 6; Fig. 7; Supplementary Fig. 7**), making unlikely that MLL1 protein is still present after tamoxifen treatment in *Pax7^{CE/+};Mll1^{+/+};Rosa^{YFP}* mice.

Minor Comments:

1. In Fig. 1 or Fig. 3, the Myf5 protein levels should be examined by western blot analyses in the Mll1 cKO conditions.

Previously, the rabbit anti-Myf5 antibody from Santa Cruz (sc-302) was used to detect Myf5 protein by Western blot and immunostaining (McKinnel *et al.*, 2008, PMID: 18066051; Chang *et al.*, 2018; PMID: 29681515). However, this antibody has been discontinued and we haven't found another anti-Myf5 antibody that reliably detects My5 protein.

2. There are multiple corrections in the text that are needed and have been outlined. This manuscript needs to be carefully reviewed and rewritten.

We have carefully reviewed the manuscript and corrected throughout. If there are specific corrections that the reviewer would like to add, we will make the modifications accordingly.

3. To better reflect the differences in decreased H3K4me3 deposition following MLL1 KO at the two different loci (Myf5 and Pax7), % change values should be included.

As requested, we added the % change values to the figure (see **Fig. 2c**).

4. Statistical analysis is needed for the microarray data.

We apologize to the reviewer but we cannot provide statistical analysis as the microarray was performed on only a single replicate of control and *Mll1* cKO myoblasts. However, we pooled three biological samples for each replicate in order to control for biological variability. We have now specified it in the “methods” section. Confirming the robustness of our microarray data, we performed RT-qPCR on several genes (up- and downregulated as well as non-variant) with biological replicates that revealed the same expression pattern and are statistically significant.

5. Fig. S3 is missing the graph legend (for the colored bar graphs).

The graph legend has been added.

6. The figure subpanel labeling (A, B, etc.) should be consistent in all the figures and with references in the text (upper vs. lower case).

We have made sure that the figure labeling and references are consistent.

7. Some of the methods subsections (ChIP as an example) need additional detail.

We added details in the ChIP subsection and in the following methods subsections: Muscle regeneration and histology; Primary myoblast isolation and culture; Protein extraction, Histone extraction and Western blotting. However, we would agree adding more details if required.

Reviewer #3 (Remarks to the Author)

Addicks et al. report that deletion of MLL1, a H3K4 methyltransferase, in satellite cells resulted in reduced H3K4me3 at both Pax7 and Myf5 promoters and impaired their expression. *Mll1* cKO, but not *Mll2* cKO, satellite cells displayed proliferation defects and *MLL1*^{-/-} mice had reduced satellite cell proliferation and self-renewal and impaired skeletal muscle regeneration after injury. Pax7 overexpression partially corrected the proliferation defect and fully restored Pax7 and Myf5 expression. Overall, this is a well conducted and interesting study.

We thank the reviewer for the appreciation of our work.

1. Table 1. In *Mll1* cKO myoblasts, expression of 929 genes was increased and of 78 genes (15 coding mRNAs) decreased. Given the positive role exerted by H3K4me3 on transcription, it is likely that upregulation is an indirect effect of MLL1 deletion. ChIP-seq for MLL1 would identify direct versus indirect targets.

We mistakenly reported in the manuscript that 929 genes were up-regulated. In fact, our re-analysis of the microarray data indicates that 322 probe sets are up-regulated, corresponding to 281 annotated genes. We identified 26 annotated genes that are down-regulated, and 21 of these encode mRNAs. We apologize for this mistake, and we have made the corrections in the manuscript. To identify direct and indirect MLL1 targets, we intersected our data with publicly available MLL1 ChIP-seq data from C2C12 myoblasts (Cheng *et al.*, 2014, PMID: 24656132). We determined that 9 out of the 21 down-regulated genes (43%) are direct MLL1 targets, whereas only 23% (64/281) of the up-regulated genes are direct MLL1

targets. Therefore, these results confirm that the up-regulation of most genes observed in *Mll1* cKO myoblasts results from indirect effects of MLL1 loss. We now discuss these results in the manuscript.

2. A GO analysis and discussion of upregulated genes in *Mll1*cKO myoblasts would be helpful.

As suggested by the reviewer, we performed a GO term analysis on the upregulated genes using g:profiler, which has been now added as **Supplemental Table S2**. We identified biological process (BP) terms associated with negative regulation of cell migration, cell motility and locomotion, and cellular component (CC) terms associated to myofibril, contractile fiber and sarcomere. We added this information in the Discussion section.

3. Myogenin is not expressed in myoblasts and thus its expression is not expected to be affected by MLL1.

We agree with the reviewer that myogenin is not expressed in proliferating myoblasts at the protein level. However, knowing that myogenic differentiation coincides with Pax7 downregulation and myogenin upregulation (Bentzinger *et al.*, 2012, PMID: 22300977; Olguin *et al.*, 2012, PMID: 21615681), we initially thought that loss of Pax7 in *Mll1* cKO myoblasts was due to *myogenin* upregulation and precocious differentiation. However, our RT-qPCR data revealed that *myogenin* expression was not changed after *Mll1* deletion (**Fig. 1e**), thus ruling out the possibility that loss of Pax7 was due to precocious differentiation.

4. Figure 3. Comparable number of myoblasts are shown for control and *Mll1* cKO in Figure 3a. However, less than 20% of *Mll1* cKO myoblasts are Pax7+. What is the identity of the remaining 80%? Do they retain other myoblasts-specific markers?

Mll1 cKO myoblasts do retain myogenic competence as they are able to differentiate and fuse into myotubes. Moreover, they express *Myod1* at the transcript level (**Fig. 1e** and **Supplementary Fig. 3c**). However, to fully confirm that *Mll1* cKO myoblasts retain myogenic identity, we performed a co-immunostaining of MyoD and Syndecan 4 (Sdc4) on proliferating myoblasts (**Supplementary Fig. 3a, b**). Although *Mll1* cKO myoblasts lose Pax7 expression, they still express myoblast-specific markers such as MyoD or Sdc4. Therefore, our results confirm that Pax7-negative *Mll1* cKO cells remain myogenic.

5. Figure 4c. As noted in the Discussion, it is surprising that CMV-Pax7 expression restored H3K4me3 at *Myf5* in *Mll1* cKO myoblasts. A ChIP-qPCR for Set1A/B in *Mll1* cKO myoblasts would directly evaluate the authors' suggestion that this methyltransferase can substitute for MLL1.

In order to evaluate whether another HMT can substitute for MLL1 in the deposition of the H3K4me3 mark at the *Myf5* and *Pax7* loci, we reasoned that a decrease in H3K4me3 mark would lead to a decrease in *Myf5* and *Pax7* transcripts. We designed knockdown experiments using siRNA treatments on primary myoblasts as follows:

a) We selectively knocked down the 4 HMTs *Mll1*, *Mll2*, *Set1a* and *Set1b* (**Supplementary Fig. 4e**).

This experiment revealed that only *Mll1* knockdown leads to the downregulation of both *Pax7* and *Myf5*, suggesting that only MLL1 can regulate both *Pax7* and *Myf5* expression in primary myoblasts.

b) We knocked down *Mll1* in combination with either *Mll2* or *Set1a* or *Set1b* (**Supplementary Fig. 4f**).

The results showed that *Pax7* mRNA level was further decreased when either *Mll2* or *Set1b* was knocked down in combination with *Mll1* siRNA treatments. This suggests that MLL2 or Set1b can compensate and substitute for the loss of MLL1 in regulating *Pax7* expression. This correlates with our results obtained in

Mll1:Mll2 dcKO myoblasts (**Fig. 1c**) showing a larger decrease of Pax7 expression when both *Mll1* and *Mll2* are knockout. Consistent with Pax7 expression, *Myf5* mRNA level was also further decreased when either *Mll2* or *Set1b* was knocked down in combination with *Mll1* siRNA treatments. These results also recapitulate our observations that the decrease in *Myf5* RNA level following *Mll1* KO is more important than the *Pax7* decrease.

Nonetheless, the *in vivo* experiments performed on *Pax7^{CE/+}:Mll1^{fl/fl}:Rosa^{YFP}* mice suggest that the other HMTs cannot compensate for MLL1 in directing satellite cell survival and proliferation *in vivo*, leading to a complete impairment in muscle regeneration.

All these data have been added to the **Supplementary Fig. 4** and discussed.

6. Supplemental Figure 5. ChIP-qPCR for MLL1 at the Pax7 locus in quiescent and activated satellite cells would help in clarifying the seemingly distinct role of MLL1 in the two cell states.

Chromatin immunoprecipitation was performed in primary myoblasts as they provide, at low cost, unlimited chromatin material to perform ChIP. Satellite cells represent a very small cell population in the muscle tissue. FACS-based satellite cell isolation from all hind limb muscles of one mouse provides on average 10⁵ satellite cells, limiting the ChIP efficiency for MLL1. However, we successfully performed a ChIP-qPCR for H3K4me3 at *Pax7* and *Myf5* loci in freshly isolated quiescent and activated satellite cells from both control and tamoxifen-treated *Pax7^{CE/+}:Mll1^{+/+}:Rosa^{YFP}* mice. These data were added to the manuscript in **Supplemental Fig. 5b-d**. They confirm that MLL1 regulates Pax7 differentially in quiescent and activated satellite cells.

7. Microarray datasets of Mll1 cKO and Pax7 cKO myoblasts should be compared to uncover differences and similarities.

See Reviewer #2 Point 15 for response.

REVIEWERS' COMMENTS:

Reviewer #1 (Remarks to the Author):

The authors have addressed my comments in an appropriate manner, as well as most of the two other reviewers comments. I believe this work should be accepted for publication.

Reviewer #2 (Remarks to the Author):

The authors have properly addressed my previous comments. This revision is now recommended for publication in Nature Communications.

Reviewer #3 (Remarks to the Author):

The authors have satisfactorily addressed my comments.

We thank the reviewers for their comments and suggestions. The revised manuscript is significantly improved.

REVIEWERS' COMMENTS

Reviewer #1 (Remarks to the Author):

The authors have addressed my comments in an appropriate manner, as well as most of the two other reviewers comments. I believe this work should be accepted for publication.

Reviewer #2 (Remarks to the Author):

The authors have properly addressed my previous comments. This revision is now recommended for publication in Nature Communications.

Reviewer #3 (Remarks to the Author):

The authors have satisfactorily addressed my comments.